EMBO
Molecular Medicine

# RIPK1-mediated immunogenic cell death promotes anti-tumour immunity against soft-tissue sarcoma

Henry G Smith[1,†], Kunzah Jamal[2,†], Jasbani HS Dayal[3], Tencho Tenev[2], Joan Kyula-Currie[1], Naomi Guppy[2], Patrycja Gazinska[2], Victoria Roulstone[1], Gianmaria Liccardi[2], Emma Davies[1], Ioannis Roxanis[2,3,4,5], Alan A Melcher[6], Andrew J Hayes[7], Gareth J Inman[3,8], Kevin J Harrington[1,*,‡] & Pascal Meier[2,**,‡]

## Abstract

Drugs that mobilise the immune system against cancer are dramatically improving care for many people. Dying cancer cells play an active role in inducing anti-tumour immunity but not every form of death can elicit an immune response. Moreover, resistance to apoptosis is a major problem in cancer treatment and disease control. While the term "immunogenic cell death" is not fully defined, activation of receptor-interacting serine/threonine-protein kinase 1 (RIPK1) can induce a type of death that mobilises the immune system against cancer. However, no clinical treatment protocols have yet been established that would harness the immunogenic potential of RIPK1. Here, we report the first pre-clinical application of an *in vivo* treatment protocol for soft-tissue sarcoma that directly engages RIPK1-mediated immunogenic cell death. We find that RIPK1-mediated cell death significantly improves local disease control, increases activation of CD8[+] T cells as well as NK cells, and enhances the survival benefit of immune checkpoint blockade. Our findings warrant a clinical trial to assess the survival benefit of RIPK1-induced cell death in patients with advanced disease at limb extremities.

**Keywords** apoptosis; RIPK1; SMAC mimetics; soft-tissue sarcoma; TNF

**Subject Categories** Cancer; Immunology

## Introduction

Dying cells have an important role in the initiation of T-cell-mediated immunity (Kroemer *et al*, 2013). The cross-presentation of antigens derived from dying cells enables dendritic cells to present exogenous tissue-restricted or tumour-restricted proteins. Cell death mechanisms that are indicative of danger to the organism (e.g. inflammatory cell death triggered by pathogens) are potent inducers of immune responses (Ho *et al*, 2018). Cancer cells that die in this manner release inducible and constitutive DAMPs (damage-associated molecular patterns), which are direct upstream drivers of anti-tumour immunity (Yatim *et al*, 2017).

While the term "immunogenic cell death" is not yet fully defined, it is clear that activation of receptor-interacting serine/threonine-protein kinase 1 (RIPK1), and formation of ripoptosome complexes, can induce a type of death that maximally drives antigen cross-priming of CD8[+] T cells (Yatim *et al*, 2015; Aaes *et al*, 2016). Mechanistically, this is because the ripoptosome can trigger cell death as well as the release of chemokines and cytokines from dying cells that attract and stimulate immune cells (Wu *et al*, 2007; Yatim *et al*, 2015, 2017). Importantly, TNF-mediated cell death can also contribute to immune surveillance by cytotoxic lymphocytes (Kearney *et al*, 2018; Vredevoogd *et al*, 2019). Accordingly, in some cases tumour immune evasion can arise through loss of TNF (tumour necrosis factor-α) sensitivity, independently of perforin-mediated killing. Hence, re-sensitising cancer cells to TNF-mediated bystander killing might be a potent mechanism to kill antigen-negative tumour cells (Vredevoogd *et al*, 2019).

Activation of RIPK1 and RIPK1-mediated cell death is best studied in the context of TNF signalling. RIPK1 is a key node in the TNF

---

1  Targeted Therapy Team, The Institute of Cancer Research, London, UK
2  The Breast Cancer Now Toby Robins Research Centre, The Institute of Cancer Research, London, UK
3  Cancer Research UK Beatson Institute, Glasgow, UK
4  Division of Molecular Pathology, The Institute of Cancer Research, London, UK
5  Royal Free London NHS Foundation Trust, London, UK
6  The Translational Immunology Team, The Institute of Cancer Research, London, UK
7  The Sarcoma and Melanoma Unit, The Royal Marsden Hospital, London, UK
8  Institute of Cancer Sciences, University of Glasgow, Glasgow, UK
   *Corresponding author. Tel: +44 (0)20 7153 5326, Fax: +44 (0)20 7153 5340; E-mail: Kevin.Harrington@icr.ac.uk
   **Corresponding author. Tel: +44 (0)20 7153 5326, Fax: +44 (0)20 7153 5340; E-mail: pmeier@icr.ac.uk
   †These authors contributed equally to this work
   ‡These authors contributed equally to this work as senior authors

signal transduction pathway, actively controlling the balance between gene activation and cell death induction in the form of apoptosis and necroptosis. The decision as to whether RIPK1 unleashes pro-survival or pro-death signals depends on multiple control points (Annibaldi & Meier, 2018). In healthy cells, TNFR1 activation results in rapid post-translational modification of RIPK1 at the receptor complex, which actively suppresses the cytotoxic potential of RIPK1. While the E3-Ub ligases cIAP1, cIAP2, MIB2 and LUBAC conjugate inhibitory ubiquitin (Ub) chains to RIPK1 (Feltham & Silke, 2017; Annibaldi & Meier, 2018; Feltham et al, 2018), multiple kinases (e.g. IKK, MK2 and TBK1) inactivate RIPK1 via direct phosphorylation (Dondelinger et al, 2015, 2018; Jaco et al, 2017; Lafont et al, 2018). Together, this limits the ability of RIPK1 to form a death-inducing platform (Bertrand & Vandenabeele, 2011; Feoktistova et al, 2011; Tenev et al, 2011).

Inhibitor of APoptosis (IAP) protein family members are frequently overexpressed in cancer and contribute to tumour cell survival, chemo-resistance, disease progression and poor prognosis (LaCasse et al, 2008). Although best known for their ability to regulate caspases, IAPs also influence TNF signalling via modulation of RIPK1 and activation of NF-kB (Gyrd-Hansen & Meier, 2010). Not surprisingly, therefore, IAPs are considered as promising targets for the design of therapeutic strategies (Fulda, 2017). Pharmacological inhibition of cIAP1 and cIAP2 abrogates both ubiquitylation of RIPK1 and Ub-dependent activation of kinases, which in turn phosphorylate and inhibit RIPK1. Inhibiting cIAPs, therefore, unleashes the pro-death potential of RIPK1. Pharmacological inhibitors of cIAPs, frequently referred to as SMAC mimetics (SM), act by promoting the degradation of cIAP1 and cIAP2 (Vince et al, 2007; Bai et al, 2014; Fulda, 2015). As these cIAPs are guardians of RIPK1, this renders cells acutely sensitive to RIPK1-mediated cell death. SM promote IAP degradation by mimicking the intracellular cIAP antagonist SMAC/Diablo (Verhagen et al, 2000). Several SM are being evaluated in early clinical trials (Cong et al, 2019). For example, the monovalent SM LCL161 from Novartis completed phase II trials on triple-negative breast cancer in combination with paclitaxel (NCT01617668), and on relapsed or refractory multiple myeloma (NCT01955434). The pathologic complete response (pCR) rate after 12 weeks of therapy was 24.9%, compared to 23.4% observed for paclitaxel alone, in patients with positive gene expression signature. Currently, there are at least four phase I or II trials recruiting patients with other indications to test LCL161 in combination with other drugs (NCT02098161, NCT02649673, NCT03111992 and NCT02890069). The bivalent SM Birinapant has been advanced into several phase I or II trials as a single agent (NCT00993239, NCT01681368, NCT01486784 and NCT01681368) or in combination with gemcitabine (NCT01573780), 5-azacitidine (NCT02147873, NCT01828346) or other standard chemotherapeutics such as irinotecan or docetaxel (NCT01188499), carboplatin (NCT02 756130), conatumumab (NCT01940172) and pembrolizumab (NCT02587962). Even though SM are well tolerated, they have limited single agent efficacy in most cancers due to the low concentration of TNF in the tumour microenvironment that would drive RIPK1-dependent cell death in the absence of cIAPs (Fulda, 2015).

Although TNF can drive RIPK1-induced cell death, the premise to harness the cytotoxic properties of TNF as an anticancer therapy is limited by the severe toxicity that results upon its systemic administration (Duprez et al, 2011; Zelic et al, 2018). However, such treatment-limiting toxicities can be avoided by the locoregional administration of TNF via isolated limb perfusion (ILP), a specialised surgical treatment of extremity malignancies (Eggermont et al, 2003). Indeed, application of TNF, most commonly in combination with melphalan (Mel), represents the current standard-of-care (SOC) treatment for targeting cancers in the extremities that are unsuitable for surgical resection, such as extremity soft-tissue sarcomas (ESTS, which represent 40% of soft-tissue sarcomas), squamous cell carcinoma (SCC), Merkel cell carcinoma (MCC), in-transit melanoma and fibromatosis (Smith & Hayes, 2016). Sarcomas account for > 20% of all paediatric solid malignant cancers and < 1% of all adult solid malignant cancers. Soft-tissue sarcomas are the most frequent, with fibrosarcomas accounting for roughly 7% of all sarcomas. During the procedure of isolated limb perfusion, TNF is used for its vasodilating properties, allowing enhanced uptake of Mel by the tumour (Deroose et al, 2011b). However, at present, the cytotoxic properties of the cytokine are not being exploited by the TNF/Mel standard-of-care ILP regimen. Clearly, improvements to the ILP standard-of-care treatment regimen are urgently needed as the duration of response is limited in the absence of any further intervention, with the majority of patients developing locally progressive disease within 12 months (Smith et al, 2015).

To evaluate the possibility of harnessing RIPK1's cytotoxic potential during ILP, we combined the current standard-of-care treatment regimen (ILP-TNF/Mel) with pharmacological inhibitors of IAPs (SM) and evaluated its efficacy in an immune-competent rat model of extremity sarcoma (ESTS). Our data demonstrate that the combination of TNF/Mel with SM mediates RIPK1-dependent cell death, which significantly improves local disease control. Mechanistically, RIPK1-mediated cell death increases activation of infiltrating CD8[+] T cells and NK cells and enhances the survival benefit of immune checkpoint blockage. Our work warrants clinical evaluation in patients with ESTS.

## Results

### Combination therapy with SM results in Ripk1-assisted cell death following standard-of-care TNF/Mel treatment

Treatment with TNF and melphalan via isolated limb perfusion (ILP) represents a standard-of-care treatment protocol for advanced extremity malignancies unsuitable for surgical resection (Lienard et al, 1992; Eggermont et al, 2003). Although 60% of patients have a significant response to ILP-TNF/Mel (Smith & Hayes, 2016), the duration of response is limited with median progression-free survival limited to 12 months (Smith et al, 2015). Therefore, novel agents or treatment combinations are urgently needed to improve outcome.

To evaluate the possibility of harnessing RIPK1's cytotoxic and immunogenic potential during ILP-based and TNF-mediated treatments in combination with SM, we employed a syngeneic, immune-competent rat model of ILP-TNF/Mel using the syngeneic BN175 sarcoma cell line in Brown Norway rats (Pencavel et al, 2015; Wilkinson et al, 2016). Due to the complex micro-surgical procedures involved during ILP, in vivo investigation of this technique requires the use of a rat rather than a mouse model. Importantly, this rat model closely resembles the clinical scenario seen in many

patients with advanced limb sarcomas after treatment with standard ILP-TNF/Mel, where an initial local response is followed by local disease progression that may occur before the development of metastatic disease (Pencavel *et al*, 2015; Wilkinson *et al*, 2016; Smith *et al*, 2019). Prior to *in vivo* application, we first evaluated the death pathways that are activated in BN175 cells upon treatment with various combinations of TNF, Mel and SM in an *in vitro* setting.

While the standard-of-care treatment TNF/Mel reduced cell viability of BN175 cells only at later time points (48 h), the addition of SM to TNF/Mel potently killed these cells at an early time point (24 h; Fig 1A, left panel). Also, at later time points, TNF/Mel/SM was more effective in killing BN175 cells than the standard-of-care treatment. Importantly, TNF/Mel/SM resulted in potent complex-II formation and caspase activation (Figs 1B–D and EV1A). In complete contrast, the standard-of-care treatment TNF/Mel did not drive formation of Ripk1:Caspase-8 (Casp-8) complexes, as judged by co-immunoprecipitation with an anti-FADD antibody and proximity ligation assay (PLA) with specific antibodies for Ripk1 and Casp-8 that successfully detect complex-II formation (Orme *et al*, 2016; Liccardi *et al*, 2018). Co-treatment with SM caused cell death that was dependent on the kinase activity of Ripk1 and Casp-8 at 24 h. This is evident as cell death was suppressed by genetic or pharmacological inhibition of Ripk1 or Casp-8 (Figs 1E, and EV1B and C). Previous work established that treatment with Mel activates the intrinsic death pathway (Matsura *et al*, 2004; Gomez-Bougie *et al*, 2005; Park *et al*, 2013). Accordingly, Mel and TNF/Mel treatment of BN175 resulted in increased caspase activity, which was rescued by co-treatment with the caspase inhibitor zVAD-fmk (Fig EV1D). Ectopic expression of BCL2 also suppressed Mel- or TNF/Mel-induced intrinsic apoptosis in HT1080 cells (Fig EV1E), corroborating that exposure to Mel triggers mitochondria-dependent apoptosis. Not surprisingly, therefore, TNF/Mel-induced cell death was not affected by loss of *Ripk1* or *Casp-8*, indicating that Mel-mediated cell death does not involve Ripk1-assisted cell death (Figs 1E, and EV1B and C).

## SM sensitises cells from human extremity malignancies to RIPK1-induced cell death

Next, we tested the sensitivity of a range of cells derived from malignancies that can be treated via ILP-TNF/Mel to TNF-induced and RIPK1-dependent cell death. Treatment with TNF resulted in the formation of the TNF receptor signalling complex-I (TNFR-SC, also referred to as complex-I) in the human fibrosarcoma cell line HT1080, as evidenced by the recruitment of *bona fide* TNFR-SC components such as RIPK1, SHARPIN and TRADD (Fig EV2A) (Micheau & Tschopp, 2003). Upon concomitant inhibition of IAPs with SM-164, TNF potently triggered formation of complex-II (Fig 2A), caspase activation (Fig EV2B) and cell death (Fig 2B) in the human fibrosarcoma cell line HT1080. Cell death was blocked by co-treatment with the caspase inhibitor zVAD-FMK (Fig 2B) or *RIPK1, CASP-8* depletion (Fig 2C), indicating that TNF/SM exclusively drives RIPK1-mediated apoptosis in these cells. Virtually identical results were obtained in A375, MeWo and DO4 melanoma cells (Figs 2D–I and EV2C–H), extending this observation to other human malignancies treated with ILP-TNF/Mel regimen. While most cancer cells are resistant to treatment with SM alone (Fig 2C, E, F and G), some cancer cells, such as the sarcoma cell line SW872, are

sensitive to single agent SM treatment (Figs 2I, and EV2C and D). Importantly, the sensitivity of such cells can be further enhanced by co-treatment with TNF (Fig 2I).

Extremity malignancies not only comprise of fibrosarcomas and melanomas but also of epithelioid sarcomas, synovial sarcoma and cutaneous squamous cell carcinoma (cSCC) (Smith & Hayes, 2016). Importantly, we found that cells from such malignancies are sensitive to TNF/SM treatment, which is caspase-dependent (Figs 2J and K, and EV2J and K). Furthermore, cSCC, which comprise 20% of all non-melanoma skin cancers, can be treated via ILP-TNF/Mel (Huis In 't Veld *et al*, 2018). In fact, ILP-TNF/Mel can achieve a limb salvage rate of 80% in such patients with a 2-year overall progression-free survival of 67%. Accordingly, cell lines derived from patients with such malignancies (Hassan *et al*, 2019) are sensitive to TNF/SM-induced cell death that is caspase-dependent (Figs 2L–N and EV2L–N). Together, our observations indicate that SM sensitises extremity malignancies to TNF-induced cell death, suggesting that it might be beneficial to incorporate SM into current ILP-TNF/Mel, standard-of-care treatment regimens.

## RIPK1-induced cell death delays tumour growth and prolongs survival

Next, we investigated whether SM enhances the efficacy of TNF/Mel under *in vivo* settings. Our *in vivo* model of ILP with BN175 sarcoma mimics the clinical scenario, with an initial response to ILP-TNF/Mel rapidly followed by local relapse (Smith *et al*, 2019). Of note, this rat sarcoma model is highly aggressive, fast proliferating, and it is difficult to achieve a response. We used the SM Birinapant (Bir) for our *in vivo* experiments as this compound has undergone clinical trial and is well tolerated (Condon *et al*, 2014; Amaravadi *et al*, 2015). Importantly, Bir and SM-164 display the same efficacy, killing BN175 cells to the same extent when combined with TNF/Mel (compare Figs 1A and EV3A).

Tumour-bearing rats underwent ILP with the indicated therapeutics and were then maintained for tumour growth and survival endpoints (Figs 3A and B, and EV3B–E). While the delivery of SM as a single therapy had minimal efficacy in comparison to untreated controls, the combination of TNF/Mel with SM (red line), delivered by ILP, significantly delayed tumour growth when compared to TNF/Mel alone. The delay in tumour growth of animals treated with TNF/Mel/SM translated into significantly prolonged survival when compared to the standard-of-care TNF/Mel (Fig 3B, median 24.5 vs. 18 days, $P = 0.0063$, log-rank test). In clinical practice, the efficacy of ILP is commonly evaluated by the percentage of non-viable cells (dead cells, generally referred to as "necrosis") in a resected specimen. To investigate the impact of TNF/Mel/SM, tumours were collected 4 days post-ILP and subjected to histopathological analysis (Fig 3C). These analyses indicated that TNF/Mel/SM significantly increased tumour death compared to TNF/Mel alone ($P = 0.0034$, unpaired *t*-test) (Fig 3C), in keeping with augmented therapy in this therapeutic cohort.

Even though TNF/Mel/SM treatment significantly prolonged survival in this highly aggressive model, tumour-bearing animals still eventually reached the tumour growth endpoint (Fig EV3B–E). To test whether this was due to the development of resistance to the TNF/Mel/SM treatment or was caused by attaining a transient state called "persistence" without mutation (Hangauer *et al*, 2017;

Viswanathan *et al*, 2017), we isolated drug-tolerant tumour cells from three rats (124, 133 and 136) that developed sarcomas post-treatment. When these tumour cells were re-challenged, they were as sensitive to TNF/Mel/SM treatment as the parental BN175 cells

(Fig 3D and E). This indicates that the surviving tumour cells are persister cells and that drug tolerance was not due to clonal evolution of resistance mechanisms. This is in agreement with the notion that ILP-mediated treatment is a one-off treatment protocol and so

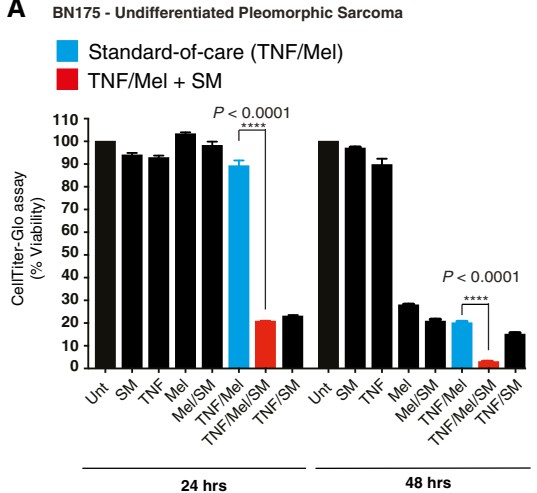

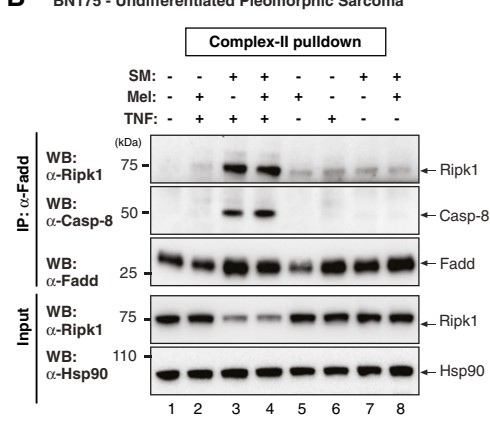

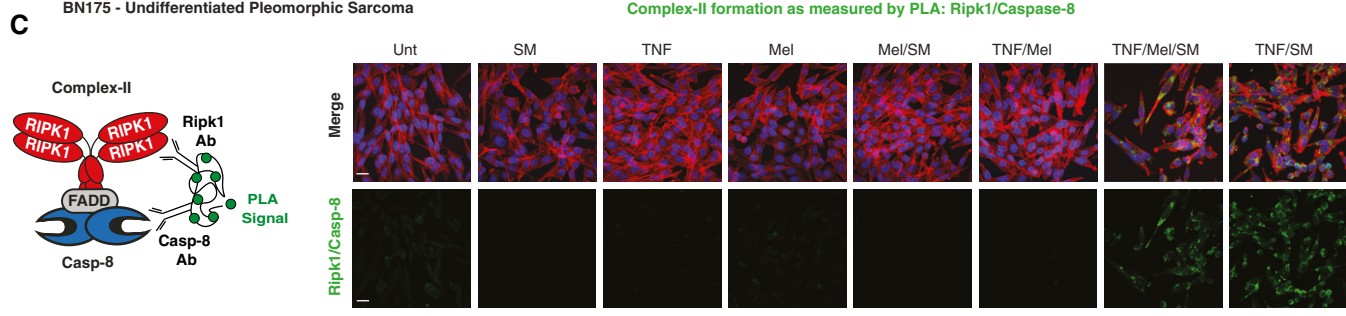

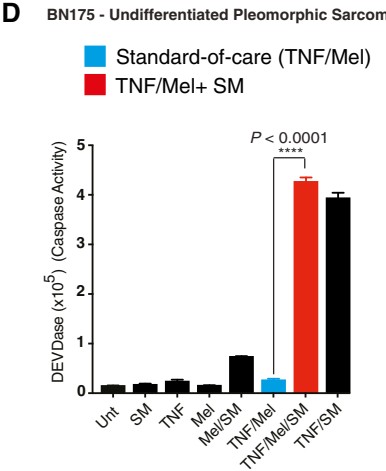

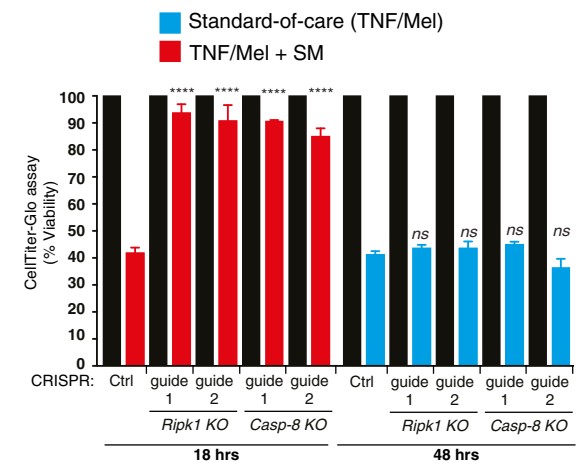

**Figure 1.**

**Figure 1.  The addition of SM to TNF/Mel sensitises cells to RIPK1-dependent cell death.**

A   Cell viability analysis using CellTiter-Glo of BN175 cells treated with the indicated agents for 24 and 48 h (*n* = 4 biological replicates). Cell viability values are displayed as a percentage of the relative untreated control. DMSO (Unt), TNF (10 ng/ml), SM (100 ng/ml) and Mel (3.3 μM). SM represents SM-164. Error bars represent SD and statistical analysis was performed with an unpaired *t*-test, ****$P \leq 0.0001$. Exact *P* values are shown in Appendix Table S1.

B   TNF-induced complex-II immunoprecipitation. BN175 cells were treated with the indicated agents for 8 h. FADD immunoprecipitation was performed followed by Western blot analysis (*n* = 3 biological replicates, shown is a representative experiment). DMSO (Unt), zVAD (10 μM), TNF (10 ng/ml), SM (100 ng/ml) and Mel (3.3 μM).

C   Proximity ligation assay between Ripk1 and Casp-8 performed in BN175 cells upon treatment with the indicated agents for 8 h (*n* = 3 biological replicates, shown are representative images). Scale bar represents 10 μm. DMSO (Unt), zVAD (10 μM), TNF (10 ng/ml), SM (100 ng/ml) and Mel (3.3 μM).

D   DEVDase activity assay of BN175 cells treated with the indicated drugs for 6 h (*n* = 3 biological replicates). DMSO (Unt), TNF (10 ng/ml), SM (100 ng/ml) and Mel (3.3 μM). Error bars represent SD, and statistical analysis was performed with an unpaired *t*-test, ****$P \leq 0.0001$. Exact *P* values are shown in Appendix Table S1.

E   Cell viability analysis using CellTiter-Glo of BN175 CRISPR/Cas9 *Ripk1* and *Casp-8* knockouts (KO) cells treated with the indicated agents for 18 and 48 h (*n* = 3 biological replicates). Cell viability values are displayed as a percentage of the relative untreated control. SM represents SM-164. Error bars represent SD, and a one-way *ANOVA* was performed to compare the mean value of each treatment to the treated BN175 CRISPR/Cas9 control (Ctrl), ****$P \leq 0.0001$. Exact *P* values are shown in Appendix Table S1.

Source data are available online for this figure.

persistence more likely reflects lack of exposure rather than true resistance.

The observation that surviving tumour cells are persister cells has important implication for second-line treatment regimens aimed at targeting residual disease. It indicates that the same immunogenic cell death pathway can be re-engaged. To test this principle, we activated Ripk1-induced cell death in the three drug-tolerant tumour lines (124, 133 and 136) using Riboxxol. Riboxxol is a synthetic TLR3 agonist that triggers Ripk1 activation (Lucifora *et al*, 2018; Schau *et al*, 2019) and that is well tolerated when administered systemically or via intra-tumour injections. We found that in combination with SM, Riboxxol potently killed all three drug-tolerant tumour lines (Fig 3F). This suggests that Riboxxol/SM may be used as second-line treatment regimen for patients that develop locally progressive disease after ILP (Smith *et al*, 2015).

An increase in locoregional toxicity was noted with TNF/Mel/SM when compared with TNF/Mel (Fig EV3F). Locoregional toxicity is typically graded using the Wieberdink scale (Wieberdink *et al*, 1982), with the majority of patients experiencing grade I–III reactions (Smith *et al*, 2015). Minimal toxicity was noted when tumour-bearing rats were treated with TNF/Mel, equating to grade I reactions (Fig EV3F). In contrast, TNF/Mel/SM resulted in all animals developing marked erythema and superficial desquamation, equating to grade III reactions. However, no functional compromise was noted with these reactions, which gradually resolved over 2 weeks following treatment (Fig EV3F). Such effects are seen following standard-of-care ILP-TNF/Mel in patients and are effectively managed with conservative measures in the majority of patients. Together, our results suggest that the addition of SM to standard-of-care ILP-TNF/Mel significantly improves animal survival. Our data also suggest that second-line treatments with Riboxxol/SM might further enhance efficacy.

**RIPK1-induced cell death augments tumour immune infiltration**

To determine whether Ripk1-induced cell death of BN175 sarcoma cells is capable of mobilising the immune system, we profiled infiltrating immune cell populations post-therapy. Tumours were collected from rats at day 4 post-ILP (Fig 4A). Importantly, TNF/Mel/SM treatment caused an increase in immune infiltration of BN175 sarcoma, leading to elevated levels of T cells (Fig 4B) and

natural killer (NK) cells (Fig 4C). While ILP with TNF/Mel had a negative effect on NK cell infiltration (Fig 4C), co-treatment with SM significantly rescued NK activity (Figs 4D and EV4A). Moreover, treatment with TNF/Mel/SM led to a recruitment of cytotoxic T cells when compared to TNF/Mel treatment (Figs 4E and EV4B). CD3$^+$CD8$^+$CD161$^-$ cytotoxic T cells also had considerably higher Granzyme B expression upon TNF/Mel/SM treatment (Fig 4F). Alongside these increases in immune effector cells, treatment with TNF/Mel/SM resulted in a significant reduction of CD3$^+$CD4$^+$Foxp3$^-$ T helper cells, with a corresponding increase in CD3$^+$CD4$^+$Foxp3$^+$ regulatory T cells (T-regs) (Fig 4G and H). When compared to TNF/Mel, the addition of SM also led to increased expression of the activation marker ICOS on T-reg cells (Fig 4I). Together, these data suggest that the addition of SM to the ILP standard-of-care TNF/Mel treatment activates the immune system to BN175 sarcoma, leading to activation of infiltrating T cells. However, both cytotoxic and regulatory T-cell populations appear to be activated.

We also intended to use immunocompromised rats (Fisher-F344.Cg-*Foxn1*$^{rnu}$ and hooded pigmented Hsd:RH-*Foxn1*$^{rnu}$) to evaluate whether TNF/Mel/SM promotes immunogenic cell death. However, we found that the vascular anatomy in the legs of the Fisher and hooded rats are not suitable for ILP micro-surgery. This is because these rat strains lack collateral anastomosing circulation that is necessary to sustain the viability of the limb (Fig EV4D). Therefore, we were not able to use immune-compromised rat strains to address the contribution of the immune system.

**Combination therapy with SM and checkpoint inhibitors further prolongs recurrence-free survival following ILP-mediated TNF/ Mel treatment**

Next, we investigated whether combination therapy with immune checkpoint inhibitors might further improve survival. As no therapeutic rat-specific anti-CTLA-4 antibodies are available, we first tested the cross-reactivity of the murine-specific 9H10 CTLA-4 antibody clone using a flow cytometry assay (Fig EV5A). Recombinant rat CTLA-4 protein was conjugated to flow cytometry beads and then stained with an isotype control or the murine-specific 9H10 clone. Increased fluorescence was noted with the 9H10 clone when compared with the isotype control, indicating that the mouse-specific anti-CTLA-4 antibody indeed detects rat CTLA-4

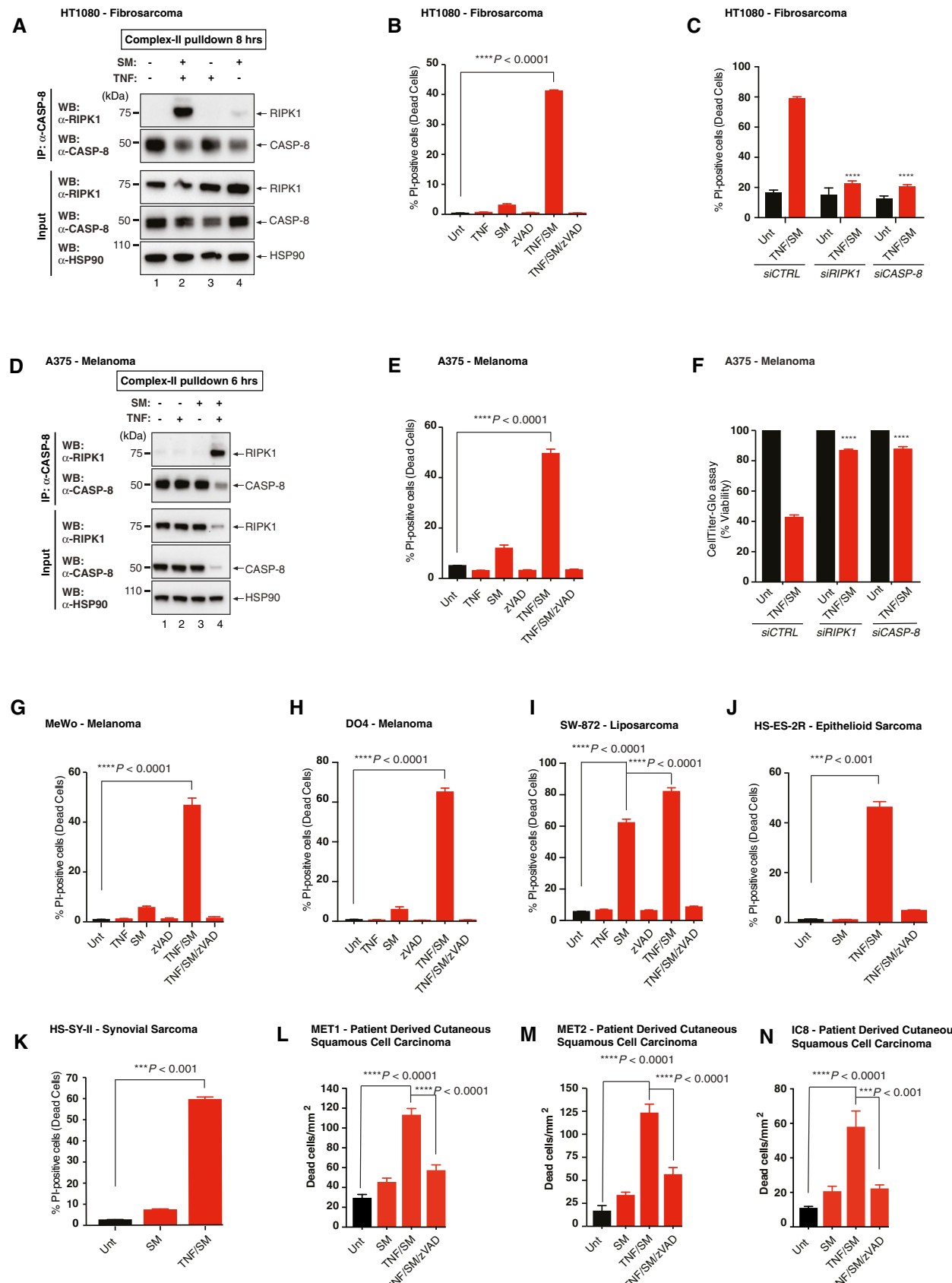

**Figure 2.**

**Figure 2.  SM sensitise a panel of human extremity malignancies cell lines to TNF-induced cell death.**

A    TNF-induced complex-II immunoprecipitation. HT1080 cells were treated with the indicated agents for 8 h. CASP-8 immunoprecipitation was performed followed by Western blot analysis ($n$ = 3 biological replicates, shown is a representative experiment). DMSO (Unt), zVAD (10 μM), TNF (10 ng/ml) and SM (100 ng/ml).

B    Cell death analysis measured by Celigo of PI-positive HT1080 cells, and cells were treated with the indicated agents for 24 h. ($n$ = 3 biological replicates). DMSO (Unt), zVAD (10 μM), TNF (10 ng/ml) and SM (100 ng/ml). SM represents SM-164. Error bars represent SD, and an unpaired $t$-test was performed, ****$P \leq 0.0001$. Exact $P$ values are shown in Appendix Table S1.

C    Cell death analysis by Celigo of PI-positive HT1080 cells treated with the indicated agents for 72 h following *siCTRL*, *siRIPK1* or *siCASP-8* knockdown for 48 h ($n$ = 3 biological replicates). DMSO (Unt), TNF (10 ng/ml) and SM (100 ng/ml). SM represents SM-164. Error bars represent SD, and a one-way *ANOVA* was performed, ****$P \leq 0.0001$. Exact $P$ values are shown in Appendix Table S1.

D    TNF-induced complex-II immunoprecipitation. A375 cells were treated with the indicated agents for 6 h. Caspase-8 immunoprecipitation was performed followed by Western blot analysis. DMSO (Unt), zVAD (10 μM), TNF (10 ng/ml) and SM (100 ng/ml) ($n$ = 3 biological replicates, shown is a representative experiment).

E    Cell death analysis by Celigo of PI-positive A375 treated with the indicated agents for 12 h. DMSO (Unt), zVAD (10 μM), TNF (10 ng/ml) and SM (100 ng/ml). SM represents SM-164. Error bars represent SD. Displayed are representative results from $n$ = 3, and statistical analysis was performed with an unpaired $t$-test, ****$P \leq 0.0001$. Exact $P$ values are shown in Appendix Table S1.

F    Cell viability analysis using CellTiter-Glo of A375 cells treated with the indicated drugs for 24 h following *siCTRL*, *siRIPK1* or *siCASP-8* knockdown for 48 h ($n$ = 3 biological replicates). Cell viability values are displayed as a percentage of the relative untreated control. DMSO (Unt), TNF (10 ng/ml) and SM (100 ng/ml). SM represents SM-164. Error bars represent SD, and a one-way *ANOVA* was performed to compare the mean value of each treatment to the treated A375 *siCTRL*, ****$P \leq 0.0001$. Exact $P$ values are shown in Appendix Table S1.

G–I   Cell death analysis by Celigo of PI-positive MeWo (G), DO4 (H) or SW-872 (I) cells. Cells were treated with the indicated agents for 24 h (MeWO and SW-872) or 12 h (DO4). DMSO (Unt), zVAD (10 μM), TNF (10 ng/ml) and SM (100 ng/ml) SM represents SM-164. Error bars represent SD. Displayed are representative results from $n$ = 3, and statistical analysis was performed with an unpaired $t$-test, ****$P \leq 0.0001$. Exact $P$ values are shown in Appendix Table S1.

J, K  Cell death analysis by Celigo of PI-positive HS-ES-2R (J) and HS-SY-II (K) cells. Cells were treated with the indicated agents for 48 h (HS-ES-2R) or 24 h (HS-SY-II). DMSO (Unt), zVAD (10 μM), TNF (100 ng/ml) and SM (100 ng/ml) SM represents SM-164. Error bars represent SD. Displayed are representative results from $n$ = 3, and statistical analysis was performed with an unpaired $t$-test, ***$P \leq 0.001$. Exact $P$ values are shown in Appendix Table S1.

L–N   Cell Death analysis by IncuCyte Zoom of CellTox Green-positive MET1 (L), MET2 (M) and IC8 (N) cells. Cells were treated with the indicated agents for 40 h. DMSO (Unt), zVAD (10 μM), TNF (10 ng/ml) and SM (100 ng/ml) SM represents SM-164. Error bars represent SD. Displayed are representative results from $n$ = 6, and statistical analysis was performed with a one-way *ANOVA*, ***$P \leq 0.001$, ****$P \leq 0.0001$. Exact $P$ values are shown in Appendix Table S1.

Source data are available online for this figure.

(Fig EV5A). The suitability of the rodent-specific anti-PD-1 antibody (J43) in rats has been demonstrated previously (Smith *et al*, 2019).

Next, we combined the ILP standard-of-care treatment (TNF/Mel) with SM and checkpoint inhibitors. Tumours were engrafted, and the ILP procedure performed at day 6 after implantation. Following the ILP procedure, tumour-bearing animals received an intraperitoneal injection of anti-CTLA-4 antibody, anti-PD-1 antibody or an isotype control, at day 8, 10 and 12 post-implantation. No further treatment was given, and animals were maintained for tumour growth and survival endpoints. Interestingly, the addition of checkpoint inhibitors was beneficial as they further prolonged survival when combined with ILP-TNF/Mel/SM (Figs 5A and B, and EV5B–G). This clearly demonstrates that the inclusion of SM and checkpoint inhibitors to the current standard-of-care treatment is highly favourable.

Addition of anti-CTLA-4 antibodies to ILP-TNF/Mel/SM led to a significant increase of T helper cells and also caused a reduction of intra-tumoural CD3$^+$CD4$^+$Foxp3$^+$ regulatory T cells (Fig 5C and D). On the other hand, treatment with anti-PD-1 antibodies caused a significant increase in active NK cells (Fig 5E). No alteration in the recruitment or activity status of cytotoxic T cells was noted (Fig EV5H–J).

## SM improves anti-tumour immunity of standard-of-care treatment

In clinical practice, ILP is often used as a neoadjuvant therapy prior to surgery. To replicate this scenario, we developed a neoadjuvant ILP model with the aim of securing local disease control (Fig 6A). This methodology also allowed us to evaluate whether the different treatment combinations have differential effects on the initiation of long-term anti-tumour immunity.

Animals underwent the ILP procedure 6 days after tumour implantation, followed by surgical resection at day 10. Animals were then observed up to 60 days after implantation for signs of local disease recurrence. All tumour resections were macroscopically complete, with no incidence of tumour spillage. We noticed a significant therapeutic improvement when SM was combined with the current standard-of-care treatment (TNF/Mel), with TNF/Mel/SM being significantly more effective than TNF/Mel in shrinking tumour volume at the time of surgery ($P \leq 0.0001$, unpaired $t$-test with Welch's correction) (Fig 6B). ILP-TNF/Mel/SM was also more potent than the standard-of-care treatment in limiting local recurrence, with local recurrence occurring in only one out of six animals (16.7%) treated with ILP-TNF/Mel/SM. In contrast, three out of six ILP-TNF/Mel-treated animals (50%) suffered local recurrence (Fig 6C). We also investigated the impact of checkpoint inhibitors in this setting. We noticed no further improvement in local disease control in comparison to ILP-TNF/Mel/SM alone.

Next, we investigated if ILP-TNF/Mel/SM had any impact on long-lasting anti-tumour immunity. To this end, we re-challenged animals whose tumour was resected at day 10, and which remained tumour-free till day 60 (see scheme Fig 6A). Tumours were implanted in the contralateral limb of tumour-free animals, and animals observed for tumour growth. In clinical practice, ILP is generally considered to have minimal impact on anti-tumour immunity and few if any abscopal effects. Consistent with the clinical finding, ILP-TNF/Mel had little effect on tumour growth at the challenge site in comparison to an untreated control (median 14 vs. 12 days, $P = 0.3173$, log-rank test) (Figs 6D and E, and EV6A and B). In contrast, animals that were co-treated with SM (ILP-TNF/Mel/SM) were protected from tumour challenge to a greater degree, displaying prolonged

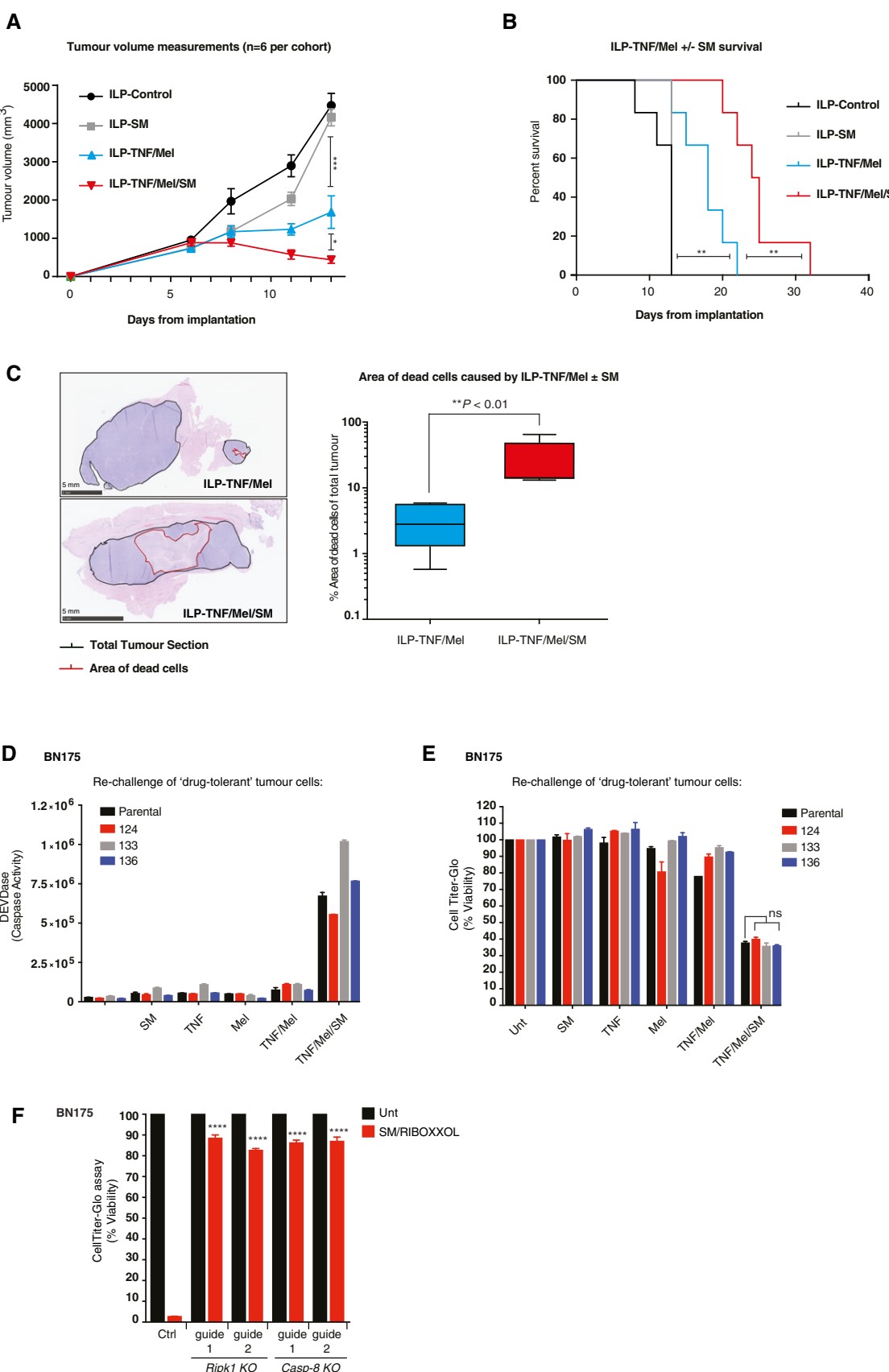

**Figure 3.**

**Figure 3.  RIPK1-induced cell death delays tumour growth and prolongs survival.**

A   Measurements of tumour volume of the respective cohorts (*n* = 6 animals per cohort). SM represents Birinapant. Error bars represent SD. Statistical analysis was performed with an *unpaired t*-test, \**P* ≤ 0.05, \*\*\**P* ≤ 0.001. Exact *P* values are shown in Appendix Table S1.

B   The addition of SM to the standard-of-care treatment ILP-TNF/Mel significantly prolonged survival compared to either modality alone (*n* = 6 rats per cohort). Statistical analysis was performed with a *log-rank test*, \*\**P* ≤ 0.01. Exact *P* values are shown in Appendix Table S1.

C   Representative images of area of dead cells and respective quantification from tumours resected from animals 10 days post ILP-TNF/Mel or ILP-TNF/Mel/SM (*n* = 6 animals per cohort). SM represents Birinapant. Median TNF/Mel is 0.4472 and TNF/Mel/SM is 1.154. TNF/Mel 25th percentile limit 0.02887 and 75th percentile limit 0.7627. TNF/Mel/SM 25th percentile limit 1.136 and 75th percentile limit 1.67. TNF/Mel Min error bar −0.2389 and Max 0.7687. TNF/SM/Mel Min error bar 1.12 and Max 1.813. Error bars represent SD. Statistical analysis was performed with an *unpaired t*-test, \*\**P* ≤ 0.01. Exact *P* values are shown in Appendix Table S1.

D   DEVDase activity assay of parental BN175 cells and drug-tolerant tumour cell lines isolated from ILP-TNF/Mel/SM-treated animals (124, 133, 136), treated with the indicated drugs for 6 h (*n* = 4 biological repeats). DMSO (Unt), Mel (3.3 μM), TNF (10 ng/ml) and SM (100 ng/ml).

E   Cell viability analysis using CellTiter-Glo of BN175 parental and 124, 133 and 136 cells treated with the indicated drugs for 24 h (*n* = 3 biological repeats). DMSO (Unt), Mel (3.3 μM), TNF (10 ng/ml) and SM (100 ng/ml). SM represents SM-164. Error bars represent SD. Statistical analysis was performed with a one-way *ANOVA*, ns = not significant. Exact *P* values are shown in Appendix Table S1.

F   Cell viability analysis using CellTiter-Glo of BN175 CRISPR/Cas9 *Ripk1* and *Casp-8* KO cells treated with the indicated agents for 24 h (*n* = 3 biological replicates). Cell viability values are displayed as a percentage of the relative untreated control. DMSO (Unt), SM (100 ng/ml) and Riboxxol (1 μg/ml). SM represents SM-164. Error bars represent SD and a one-way *ANOVA* was performed to compare the mean value of each treatment to the treated BN175 CRISPR/Cas9 control (Ctrl), \*\*\*\**P* ≤ 0.0001. Exact *P* values are shown in Appendix Table S1.

Source data are available online for this figure.

survival even in a highly aggressive model (median 20 vs. 14 days, *P* = 0.0259, log-rank test) (Fig 6D and E). Our data are consistent with the notion that treatment with ILP-TNF/Mel/SM drives anti-tumour immunity and is significantly more effective than the current standard-of-care treatment. Therefore, our findings warrant a clinical trial to assess the survival benefit of RIPK1-induced cell death in patients with advanced disease at limb extremities.

## Discussion

The pathways that initiate and execute immunogenic cell death are complex, genetically encoded and subject to significant regulation. Here, we report the first clinical application of an *in vivo* treatment protocol that directly engages RIPK1-mediated immunogenic cell death. Using a rat model of aggressive extremity sarcoma, we demonstrate that TNF-mediated cell death significantly improves local disease control, increases activation of infiltrating CD8[+] T cells and NK cells, and enhances the survival benefit of immune checkpoint blockade. Therefore, our findings warrant clinical evaluation of the survival benefit of TNF/Mel/SM in combination with checkpoint inhibitors in patients with advanced disease at limb extremities.

Regional chemotherapy by isolated limb perfusion (ILP) was introduced as a therapy to treat advanced extremity malignancies either as a standalone treatment or as a neoadjuvant treatment to downsize a malignancy and facilitate function-preserving surgery. Although combination treatment with TNF and melphalan-based ILP can provide initial good responses, the effects are generally short-lived with the disease progressing with 12 months (Eggermont *et al*, 1996, 2003; Deroose *et al*, 2012). While TNF is used for its vasodilating properties during ILP, its anticancer potential has never been exploited. SM compounds that antagonise cIAP1/2 have received attention as potential cancer therapeutics because cIAP1/2 are overexpressed in many cancer types and their elimination from cancer cells often induces TNF-driven apoptosis (Vince *et al*, 2007; Bai *et al*, 2014; Fulda, 2015).

Several lines of evidence indicate that combining SM with the current standard-of-care ILP-TNF/Mel regimen is more efficacious

in treating extremity sarcomas than ILP-TNF/Mel alone. First, treatment with SM enhances tumour kill in response to TNF/Mel *in vitro* as well as *in vivo*. Intriguingly, the area of necrotic (dead material) tumour mass post-ILP treatment is a recognised prognostic marker for local recurrence (Deroose *et al*, 2011a). Hence, the degree of dead material ("necrotic area") is used to determine whether a patient will have to undergo adjuvant radiotherapy. In line with this clinical assessment, we found that the enhanced necrotic area of ILP-TNF/Mel/SM-treated tumours correlated with improved long-term local disease control, when compared to standard-of-care ILP-TNF/Mel (in our neoadjuvant protocol). Importantly, the addition of SM to ILP-TNF/Mel may not only prolong local disease control but also reduce the requirement for radiotherapy, which is of particular value given that ILP is often performed in relatively younger patients. Second, SM compounds unleash the cytotoxic potential of TNF, driving TNF signalling in favour of RIPK1- and CASP-8-mediated cell death. This death pathway runs in parallel to the one triggered by Mel and hence adds to the complexity of cell death signalling. Third, TNF/Mel/SM-induced cell death appears to be more immunogenic, as cancer cells that die in this way more efficiently induce anti-tumour immunity. This observation is supported by recent studies demonstrating that activation of innate immune pathways within dying cells, such as activation of NF-κB, represents an earlier immunological event that regulates the outcome of T-cell priming (Yatim *et al*, 2015, 2017). This is because TNF can trigger cell death as well as NF-κB-dependent production and release of "danger signals" from dying cells that attract and stimulate immune cells. Thus, enhancing TNF killing by inhibiting IAPs is likely to drive an immunological response. Consistent with this notion, we found that the addition of SM to the standard-of-care treatment ILP-TNF/Mel leads to an increase in the number of activated CD8[+] T cells and NK cells. However, this was accompanied by an increase in the population of CD3[+]CD4[+]Foxp3[+] T-reg cells with immunosuppressive activities. Consistent with the notion that immune suppressive activities are present, we found that co-treatment with anti-CTLA-4 or anti-PD-1 antibodies significantly enhanced the survival benefit of ILP-TNF/Mel/SM. Accordingly, checkpoint blockade caused T-reg depletion or boosted NK cell activation.

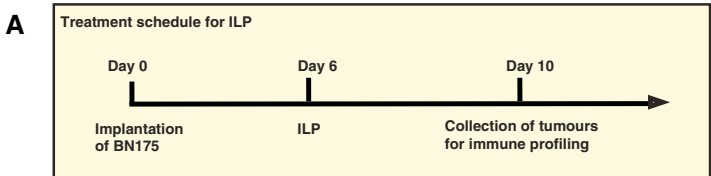

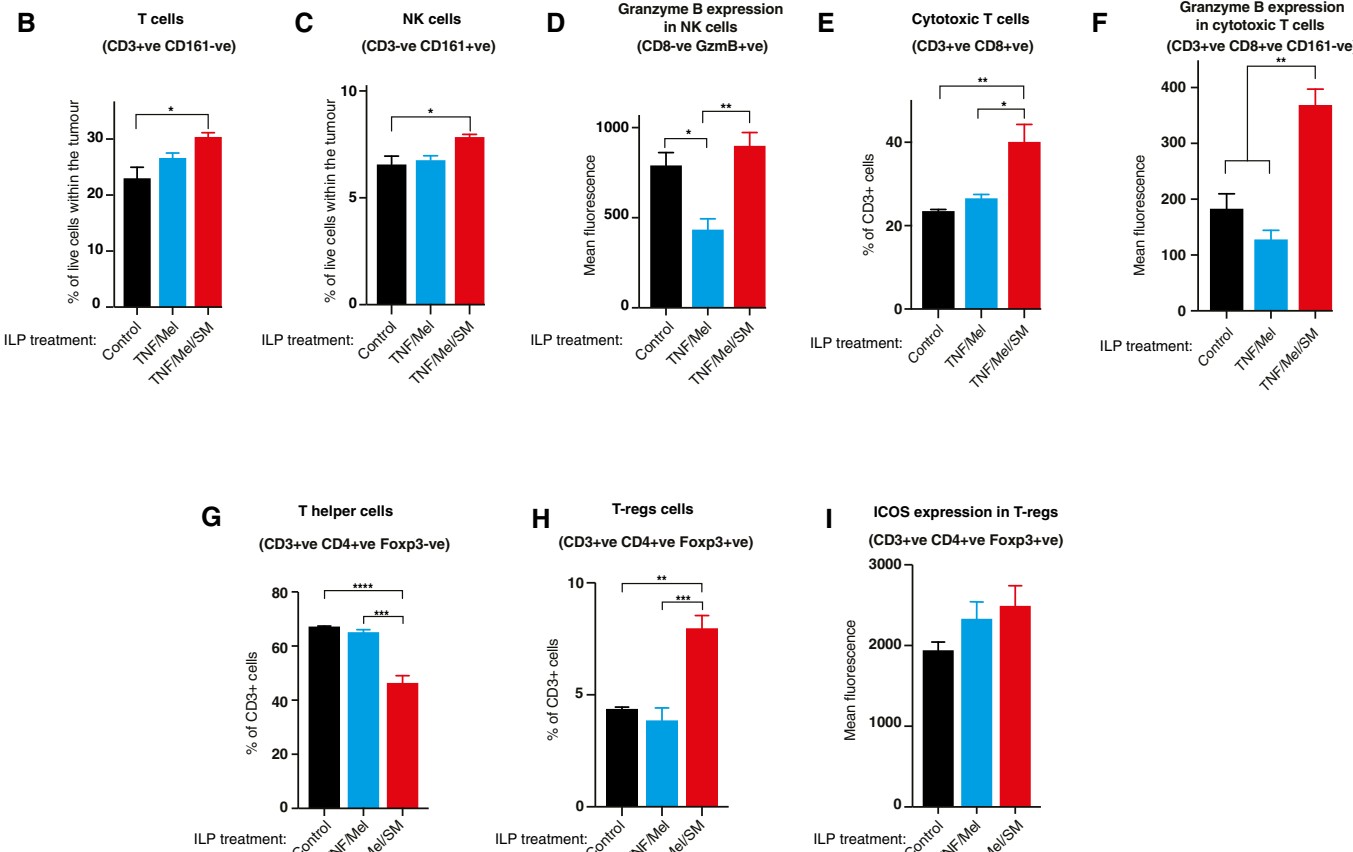

**Figure 4. RIPK1-induced cell death augments tumour immune infiltration.**

A–I Flow cytometry analysis of tumours collected on Day 10 following treatment ($n$ = 3 animals per group). (A) Schematic representation of the experimental setup. ILP with TNF/Mel/SM significantly increased infiltration by CD3$^+$ lymphocytes (B) and CD3$^-$CD161$^+$ NK cells (C). While ILP treatment with TNF/Mel significantly reduced Granzyme B-positive NK cells (D), the numbers of Granzyme B-positive NK cells were restored to normal levels upon co-treatment with SM (TNF/Mel/SM). Importantly, ILP-mediated treatment with TNF/Mel/SM significantly increased the numbers of CD3$^+$CD8$^+$ cytotoxic T cells (E). ILP-TNF/Mel/SM significantly upregulated Granzyme B in CD3$^+$CD8$^+$CD161$^-$ cytotoxic T cells (F). ILP-TNF/Mel/SM significantly reduced infiltration by CD3$^+$CD4$^+$Foxp3$^-$ T cells while (G) increasing infiltrations by CD3$^+$CD4$^+$Foxp3$^+$ T regulatory cells (H). A non-significant increase in T regulatory cell ICOS expression was also noted (I). SM represents Birinapant. Error bars represent SD and a one-way *ANOVA* was performed *$P \le 0.05$; **$P \le 0.01$; ***$P \le 0.001$; ****$P \le 0.0001$. Abbreviations: positive (+ve), negative (−ve), Granzyme B (GzmB). Exact $P$ values are shown in Appendix Table S1.

Source data are available online for this figure.

The immune cell types characterised in this study represent just a fraction of the diverse populations that comprise the tumour microenvironment, with other cell types such as macrophages, fibroblasts and B cells, as well as stromal components, increasingly recognised as crucial players in tumour response and resistance to therapy (Chen & Mellman, 2017). While it would have been ideal to investigate the contributions of such cells types in this model, the availability of high quality rat-specific reagents was a limiting factor. However, clinical translation of this combination therapy would provide the opportunity for a more comprehensive evaluation of the effects of ILP-TNF/Mel/SM on the tumour microenvironment.

Further, our data demonstrate that the addition of SM to the standard-of-care ILP treatment TNF/Mel can promote abscopal

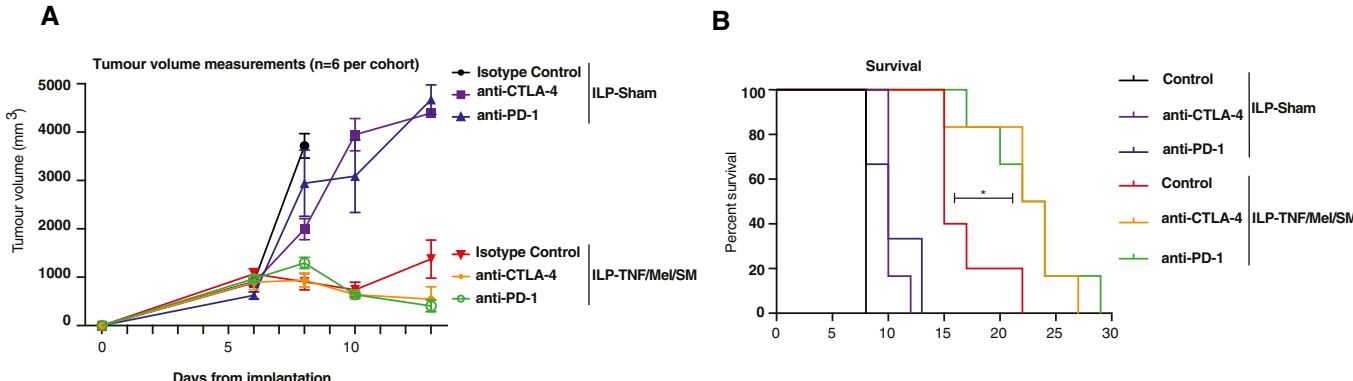

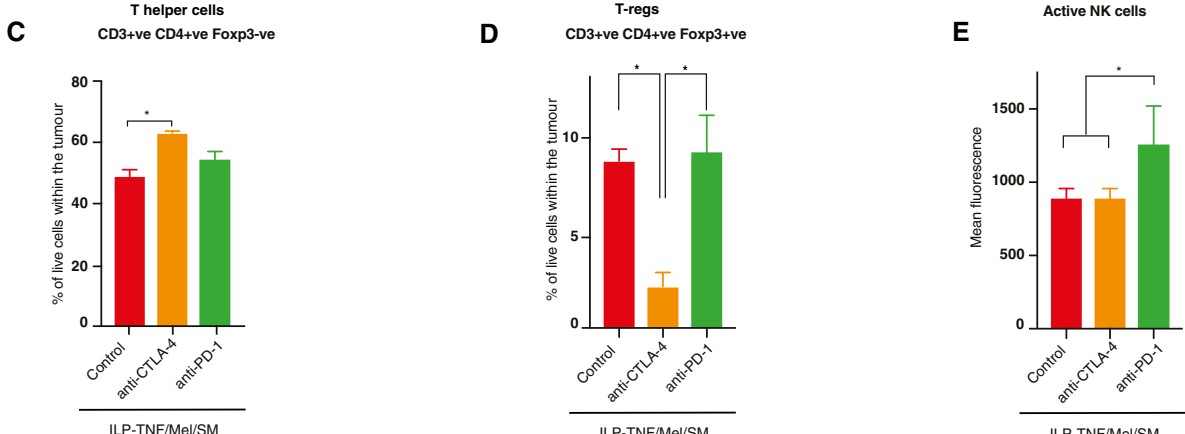

**Figure 5. Combination therapy with SM and checkpoint inhibitors further prolongs recurrence-free survival following ILP-mediated TNF/Mel treatment.**

A Measurements of tumour volume of the respective cohorts ($n$ = 6 animals per cohort). SM represents Birinapant. Error bars represent SD.

B The addition of either anti-CTLA-4 or anti-PD-1 antibodies prolonged survival in comparison to ILP-TNF/Mel/SM alone ($n$ = 6 animals per cohort). SM represents Birinapant. Statistical analysis was performed with a *log-rank test*, *$P \leq 0.05$. Exact $P$ values are shown in Appendix Table S1.

C–E Tumours were collected 4 days after the ILP procedure and analysed by flow cytometry ($n$ = 3 per cohort). The addition of anti-CTLA-4 antibodies led to a significant increase in $CD4^+Foxp3^-$ helper T cells (C) accompanied by a depletion of $CD4^+Foxp3^+$ regulatory T cells (D). The addition of anti-PD-1 antibodies led to an increase in Granzyme B (GzmB) expression on NK cells (E). SM represents Birinapant. Error bars represent SD and a one-way *ANOVA* was performed *$P \leq 0.05$. Abbreviations: positive (+ve), negative (−ve), Granzyme B (GzmB). Exact $P$ values are shown in Appendix Table S1.

Source data are available online for this figure.

immunity. Accordingly, tumour growth upon re-challenge of long-term survivors following neoadjuvant ILP-TNF/Mel/SM was significantly delayed in comparison to survivors of the ILP-TNF/Mel treatment arm (Fig 6E), indicating the induction of a degree of systemic anti-tumour immunity. This is an important issue as current treatment regimens can only treat the perfusion field and have no significant impact on the dissemination of disease or overall survival.

Even though TNF/Mel/SM treatment significantly prolonged survival in this highly aggressive model, tumour-bearing animals still eventually reached the tumour growth endpoint. This could be due to the constraint of the ILP procedure, which only allows the chemotherapy to be administered once, and this could limit the

degree of anti-tumour immunity established. Accordingly, isolated drug-tolerant tumour cells, isolated from rats that developed sarcomas post-treatment, remained as sensitive to the treatment as parental cells, demonstrating that "resistance" to the TNF/Mel/SM treatment was caused by attaining a transient state called "persistence" without mutation (Hangauer *et al*, 2017; Viswanathan *et al*, 2017). This has important ramifications, as it suggests that a similar line treatment regimen, such as SM/Riboxxol, might be used for patients that develop locally progressive disease after ILP (Smith *et al*, 2015).

The combination of SM with ILP-TNF/Mel has the potential for direct clinical translation in both palliative and neoadjuvant protocols. While our *in vivo* studies were limited to a single

sarcoma model, our *in vitro* data indicate that this treatment combination has the potential to improve response to ILP-TNF/Mel in a range of pathologies and histological subtypes. Further, while it may augment the initial response to therapy in extremity malignancies, it may also make them more susceptible to subsequent immunotherapies.

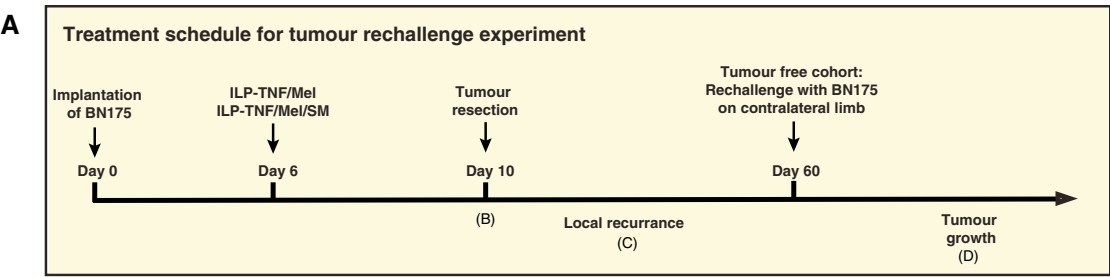

**A**  Treatment schedule for tumour rechallenge experiment

**Figure 6. SM improves anti-tumour immunity of standard-of-care treatment.**

A  Schematic representation of the experimental setup.

B  Tumour volumes at the time of surgery following ILP-TNF/Mel or ILP-TNF/Mel/SM (*n* = 6 animals per cohort). SM represents Birinapant. Error bars represent SD. Statistical analysis was performed with an unpaired *t*-test, ****$P \leq 0.0001$. Exact *P* values are shown in Appendix Table S1.

C  Individual growth curves following surgery indicated improved control following ILP-TNF/Mel/SM (*n* = 6 animals per cohort).

D  Tumour volume on re-challenge with BN175 sarcoma on contralateral limb of long-term survivors following ILP-TNF/Mel or ILP-TNF/Mel/SM (*n* = 3 animals per TNF/Mel cohort and *n* = 5 per TNF/Mel/SM cohort). Unless otherwise stated, SM represents Birinapant. Error bars represent SD.

E  ILP-TNF/Mel/SM significantly delayed growth leading to prolonged survival on re-challenge (*n* = 3 animals per TNF/Mel cohort and *n* = 5 per TNF/Mel/SM cohort). SM represents Birinapant. Statistical analysis was performed with a *log-rank test*, *$P \leq 0.05$. Exact *P* values are shown in Appendix Table S1.

Source data are available online for this figure.

# Materials and Methods

## Study design

Using an immune-competent, orthotopic and metastatic animal model of ESTS, we assessed the therapeutic benefit of adding SM to the locoregional delivery of ILP.

The ability of this combination therapy to control local and systemic disease relapse was evaluated in both palliative and neoadjuvant protocols. To elucidate the mechanism responsible for enhanced therapy, we characterised effects of treatment on the tumour microenvironment in our *in vivo* model using flow cytometry and immunohistochemistry. A single investigator performed all the surgical procedures. Sample sizes were determined using a power calculation based on detecting a 20% difference between control and treated cohorts.

## Reagents and antibodies

DISC lysis buffer (20 mM Tris–HCL pH7.5, 150 mM NaCl, 2 mM EDTA, 1% Triton X-100, 10% Glycerol, $H_2O$). The following reagents were used: human and mouse TNF (Enzo), zVAD-FMK (Apex Bio), SM-164 (gift from Shaomeng Wang), Ripk1i (GSK'963, gift from GlaxoSmithKline plc.), doxycycline (DOX) (BD Bioscience), Riboxxol (Riboxx Gmbh). The following antibodies were used: α-RIPK1 (Cell Signaling; 3493), α-CYLD (Cell Signaling; D1A10), α-SHARPIN (Proteintech), α-TRADD (BD Biosciences), α-TNFR1 [H5] (Santa Cruz Biotechnology), α-HSP90 (Santa Cruz Biotechnology), α-CASP-8—for Western blot (WB)—post-immune-precipitation (IP) (MBL), α-CASP-8—for IP [C-20] (Santa Cruz Biotechnology), α-Casp-8 for rat (Cell Signalling; 9429), α- FADD for IP and WB [M-19] (Santa Cruz Biotechnology), α-Ripk3 (Proscience; 2283). All antibodies were used at a 1:1,000 dilution.

## Tissue culture

HT1080, A375, SW-872, MeWO, DO4, HS-ES-2R, HS-SY-II, HS-ES-2M, HS-ES-I, SCC-9 and SCC-12 cell lines were obtained from ATCC. BN175 cells were a kind gift from Professor A Eggermont. HT1080, A375, SW-872, MeWO, DO4, HS-ES-2R, HS-SY-II, HS-ES-2M, HS-ES-I, SCC-9, SCC-12, BN175 and primary mouse dermal fibroblasts (MDF) cells were cultured in Dulbecco's modified Eagle's medium (DMEM). Culture media were supplemented with 10% foetal bovine serum (Gibco), penicillin and streptomycin. HT1080 cells were stably transduced with Dox-inducible BCL2 or RFP and cultured in Dulbecco's modified Eagle's medium (DMEM). Culture media were supplemented with 10% foetal bovine serum (Gibco), penicillin and streptomycin and Dox (100 ng/ml). All cells were cultured under 10% $CO_2$. Ethical approval and written informed consent was obtained from all subjects prior to isolation of primary keratinocytes from cSCC tumours as published previously (Purdie *et al*, 2011; Watt *et al*, 2011). Experiments conformed to the principles set out in the WMA Declaration of Helsinki and the Department of Health and Human Services Belmont Report. SCCMET1, SCCMET2 and SCCMET4 were isolated from the same patient from the primary, recurrence and metastatic tumour sites, respectively (Proby *et al*, 2000). SCCIC8 was isolated from an immune-competent patient and SCCT1 from a transplant patient.

## IncuCyte cell death assay

Primary keratinocytes were seeded at $1 \times 10^3$ cells/well in a 96-well plate. Cells were treated with the indicated treatments, and 0.04 μl of CellTox™ Green (Promega, G8731) express reagent was added to each well at the start of the assay. Cells were imaged every 4 h using the IncuCyte ZOOM® live cell imaging system (Essen Bioscience, Inc.) to record percentage confluence and number of dead cells (green object count/mm$^2$) per well.

## Cell death (Celigo) and viability (CellTiter-Glo)

For CeligoS assays, $8 \times 10^3$ cells were seeded in 96-well plates, and 24 h later, cells were treated as indicated for the indicated times. Hoechst (0.5 μg/ml) and PI (1 μg/ml) were added, and the percentage of dead cells was measured using the CeligoS image cytometer (Nexcelom Bioscience). Cell viability was determined using Cell-Titer-Glo reagent (Promega). Luminescence values were normalised to the median of the per-plate DMSO vehicle only control well.

## Caspase activity assays (DEVDase)

DEVDase assay was performed as previously described (Feltham *et al*, 2018). In brief, cells were plated in 96-well plates and retro siRNA transfection was performed for 40 h. After treatment, medium was removed and 1% DISC lysis buffer was added to each well. Plates were placed at −80°C to aid cell lysis. Plates were thawed at room temperature for 15 min, after which DEVDase assay mix was added to each well (NB: cell lysates were not cleared). The plates were wrapped in foil, and the reaction was incubated at room temperature for up to 24 h. DEVDase activity was read at 380 nM excitation/460 nM emission.

## Complex-I/II purification

Complex-I/II purification was essentially performed as previously described (Feltham *et al*, 2018). In brief, cells were seeded in 15 cm dishes and treated as indicated using pre-warmed media containing 3xFLAG-hTNF (0.8 μg/ml). After stimulation, media was removed and plates were washed with ice cold PBS to stop stimulation and frozen at −80°C. Plates were thawed, and cells were lysed in DISC lysis buffer supplemented with protease inhibitors and PR619 (10 μM). Cell lysates were rotated at 4°C for 20 min then clarified at 4°C at 16,873 *g* for 10 min. 20 μl of anti-FLAG M2 beads (SIGMA) were rotated with cleared protein lysates overnight at 4°C. 0 h sample: 0.8 μg/ml of FLAG-TNF was added post-lysis. 4× washes in DISC buffer with PR619 (10 μM) were performed, and samples eluted by boiling in 50 μl 1× SDS loading dye. For complex-II purification, cells were seeded in 10 cm dishes and treated as indicated in figure legends. Cells were lysed on ice as above. Cell lysates were rotated at 4°C for 20 min and then clarified at 4°C at 14,000 rpm for 10 min. 20 μl of protein G sepharose (SIGMA) with CASP-8 (C20) antibody (Santa Cruz Biotechnology) (for Human) or FADD antibody (Santa Cruz Biotechnology) (for rat) (1.5 μg antibody/mg protein lysate) were rotated with cleared protein lysates overnight at 4°C. 4× washes in wash buffer (50 mM Tris pH 7.5, 150 mM NaCl, 0.1% Triton X-100, and 5% glycerol) were performed, and samples eluted by boiling in 50 μl 1× SDS loading dye.

## Immunoblotting

Proteins lysates were quantified before separating samples by SDS–PAGE using NuPAGE Novex 4–12% Bis-Tris 1.0 mm 12 well precast protein gels (Invitrogen) in MES buffer.

## Proximity ligation assay

PLA was performed as described previously (Orme *et al*, 2016; Liccardi *et al*, 2018) using the Duolink Detection Kit (SIGMA) following manufacturer's instructions. Cells were examined with a confocal microscope (objective × 40, Zeiss LSM 710). Briefly, $1 \times 10^5$ cells were plated on a 13 mm glass coverslip (VWR). The following day, cells were treated with the indicated agents and fixed in 4% PFA (paraformaldehyde) for 10 min. Cells were permeabilised in $1 \times$ PBS, 0.5% Triton X 100 (Sigma) for 10 min and blocked for 1 h in $1 \times$ PBS, 5% BSA. The required primary antibodies were then added overnight at 4°C. Proximity ligation was then performed according to manufacturer's instructions.

## Generation of CRISPR cell lines

Guide RNAs targeting rat *Ripk1* and *Casp-8* were designed using http://crispor.tefor.net. All guides were cloned into a lentiviral vector-lentiCRISPR as previously described (McComb *et al*, 2016). Two weeks after infection, cells were FACS sorted and used for further assays. Guide sequences are as follows: *Ripk1* guide 1—CGATCGAGTGGTGAAGCTAC, *Ripk1* guide 2—TGTGAAAGTCACGGTCAACG, *Casp-8* guide 1—TACGATATTGCTGAACGTCT and *Casp-8* guide 2—GCAGATCCCGCCGACTGATA.

## Therapeutic agents

Melphalan (Alkeran) was purchased from Laboratories Genopharm, France. Recombinant hTNF was purchased from First Link Ltd (Birmingham, UK). Birinapant was purchased from LC laboratories, USA. Anti-CTLA antibody (clone 9H10) was purchased from 2BScientific (Upper Heyford, UK).

## *In vitro* validation of anti-CTLA-4 and anti-PD-1 antibody cross-reactivity

Recombinant rat CTLA-4 protein was purchased from Sino Biological (Wayne, PA, USA). After reconstitution, the recombinant protein was conjugated to flow cytometry beads using the Functional Bead Conjugation kit from Becton Dickinson (Wokingham, UK). Samples were stained with the 9H10 anti-CTLA-4 antibody and the relevant isotype control then with an Alexa Fluor 488 fluorescent secondary antibody (Thermo Fisher Scientific, Hemel Hempstead, UK) and assessed using flow cytometry. Recombinant rat PD-1 protein was purchased from Sino Biological (Wyne, PA, USA), and cross-reactivity has been previously validated (Smith *et al*, 2019).

## *In vivo* studies

Inbred, male Brown Norway rats weighing between 225–275 g were obtained from Envigo (Huntingdon, UK) (8/9 weeks old) and housed on a 12-h light/dark cycle with unrestricted access to food and water at all times. Tumours were established by injecting $1 \times 10^7$ BN175 cells subcutaneously into the left biceps femoris muscle. Treatment was started 6 days after tumour implantation. ILP was performed as previously described (Pencavel *et al*, 2015; Smith *et al*, 2019) with 50 μg TNF and 100 μg Mel. In brief, a hyperthermic perfusion with a target temperature of 40°C was performed via a perfusion circuit established using the femoral vessels and a desktop perfusion pump. For the relevant cohorts, SM (Birinapant) was added to the perfusion circuit at a dose of 1.25 mg (approximately 5 mg/kg). Perfusion was performed for 10 min and washout for 2 min. Where applicable, anti-CTLA-4 or anti-PD-1 antibodies were delivered by intraperitoneal injection at doses of 500 and 200 mg, respectively, in three repeated doses at 48 h intervals. The control groups underwent a sham ILP procedure to control for the impact of surgical intervention in these experiments. Where applicable, tumour resections were performed using a standardised compartmentectomy technique, whereby the tumour was removed en-bloc within the biceps femoris muscle (Smith *et al*, 2019). All tumours were resected with macroscopically clear margins (R1), and no evidence of tumour spillage was noted. R0 margins were not assessed due to the concern that a 1 mm margin in animal model would not have the same relevance as in the clinical context. For the re-challenge experiment in Fig 6, tumours were implanted in the opposite leg. Due to the isolated nature of ILP, this provided a remote site from the initial therapy in which to evaluate the abscopal effect. An Institutional Animal Care and Use Committee (IACUC) approved the *in vivo* experiments. The dosing of these compounds was determined in previous studies (Pencavel *et al*, 2015; Wilkinson *et al*, 2016; Smith *et al*, 2019). The 100 μg dose of Mel equates to a clinically relevant dose of 50 mg. The dosage for SMAC mimetics, and in particular Birinapant, has previously been established for mouse as 10 mg/kg (Lalaoui *et al*, 2016). By applying the FDA guidelines for dose conversion between species, this equates to 5 mg/kg for rats (Nair & Jacob, 2016), which is what we used in our *in vivo* ILP model.

## Flow cytometry

Tumours were stained with the following rat-specific conjugated antibodies: CD3-APC (1 μg/100 μl), CD4-FITC (0.25 μg/100 μl), CD8-PerCP (0.25 μg/100 μl), CD161-PE (0.5 μg/100 μl), ICOS-PECy7 (0.5 μg/100 μl), Granzyme B-Pacific Blue (0.5 μg/100 μl) (BioLegend, San Diego, CA, USA); Foxp3-Alexa700 (0.25 μg/100 μl), Viability-APC Cy7 (1 μl/ml) (Thermo Fisher Scientific). To determine the total numbers of NK and $CD8^+$ CTLs within the tumours, counting beads were added to each sample to allow the proportion of each sample analysed to be calculated. This allowed a multiplication factor to be calculated so that the total immune infiltrate of the tumours could be estimated. This was then divided by tumour weight.

## Immunohistochemistry

Formalin-fixed paraffin-embedded (FFPE) sections were analysed haematoxylin and eosin using standard protocols. The Nanozoomer-XR platform (Hamamatsu Photonics, Welwyn Garden City, UK) was used to obtain digital images of stained slides.

## The paper explained

### Problem

The prevention and treatment of sarcomas are areas of significant unmet need. 40% of soft-tissue sarcomas occur at the extremities (extremity soft-tissue sarcomas, ESTS). Surgical resection and radiotherapy are the cornerstones of the therapeutic strategy for the majority of patients. However, in those patients with locally advanced or metastatic disease, there are currently few effective systemic treatment options. Isolated limb perfusion (ILP) is a specialised surgical technique for delivering regional bio-chemotherapy, most commonly melphalan (Mel) in combination with tumour necrosis factor-$\alpha$ (TNF), to a tumour-bearing limb. During the procedure of ILP, TNF is used for its vasodilating properties, allowing enhanced uptake of Mel by the tumour. However, at present, the cytotoxic properties of TNF are not being exploited by the TNF/Mel standard-of-care ILP regimen. Recent data demonstrate that TNF-induced activation of receptor-interacting serine/threonine-protein kinase 1 (RIPK1), and formation of ripoptosome complexes, can induce a type of death that maximally drives antigen cross-priming of CD8[+] T cells. Moreover, TNF-mediated immunogenic cell death can also contribute to immune surveillance by cytotoxic lymphocytes. However, no clinical treatment protocols have yet been established that would exploit the cytotoxic and immunogenic potential of TNF.

### Results

In our study, we report the first application of an *in vivo* treatment protocol for soft-tissue sarcoma that directly engages RIPK1-mediated immunogenic cell death. We find that RIPK1-mediated cell death significantly improves local disease control, increases activation of CD8[+] T cells as well as NK cells and enhances the survival benefit of immune checkpoint blockade. To harness RIPK1's cytotoxic potential during ILP, we combined the current standard-of-care treatment regimen (ILP-TNF/Mel) with pharmacological inhibitors of IAPs (SMAC mimetics, SM) and investigated the potential to promote anti-tumour immune responses as well as augment response to PD-1 blockade in an animal model of extremity sarcoma.

### Impact

The finding that TNF-mediated cell death significantly improves local disease control and enhances the effect of immune checkpoint blockade warrants a clinical trial to assess the survival benefit of RIPK1-induced cell death in patients with advanced disease at limb extremities.

## Statistical analysis

All statistical analyses were conducted using GraphPad Prism, version 7.0 (San Diego, USA). Grouped data are presented as means ± SEM unless otherwise stated. Differences between multiple groups were assessed using one- or two-way ANOVA with *post hoc* analysis using Bonferroni's correction. Differences between two groups were assessed using a two-tailed unpaired *t*-test. Survival outcomes were compared using the Kaplan–Meier method and the log-rank test. Significance was determined at $P < 0.05$.

**Expanded View** for this article is available online.

## Acknowledgements

The authors are indebted to Professor A. Eggermont for the donation of the BN175 cell line and S. Wang for SM164. We also thank members of the Harrington and Meier laboratory for helpful discussions. We would like to apologise to the many authors whose work we could not cite due to space restrictions. Work in the Meier laboratory is funded by Breast Cancer Now as part of Programme Funding to the Breast Cancer Now Toby Robins Research Centre (CTR-QR14-007) and postgraduate studentships from Cancer Research UK (CRUK) (CRM089X). Work in the Harrington laboratory was funded by The Royal College of Surgeons of England (160729), Sarcoma UK (SUK203.2016/ SUK203.2017), The Meirion Thomas Cancer Research Fund, The McAlpine Foundation, Rosetrees Trust (A1292), Royal Marsden/Institute of Cancer Research and National Institute of Heath Research Biomedical Research Centre (NIHR BRC).

## Author contributions

KJ, HGS, KJH, PM designed the research. HGS performed and analysed experiments in Figs 3A, B, EV3B–F, 4A–I, EV5, 5, EV6 and 6. KJ performed and analysed experiments in Figs 1A, B, D, E, EV1B–E, 2A–K, EV2A–L, 3C–E and EV3A. JHSD performed and analysed experiments in Figs 2L–N, and EV2M and N. GL performed and analysed experiments in Figs 1C and EV1A. TT performed and analysed experiment in Fig 3F and assisted with data analysis and experimental design. JK-C, VR, and ED assisted with research. HGS, KJ, AM, AJH, KJH, GJI and PM supervised the study. KJ, KJH and PM supervised and handled the revision process. KJ, HGS, KJH, and PM wrote the manuscript. NG, PG and IR assisted with data analysis.

## Conflict of interest

The authors declare that they have no conflict of interest.

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
