## [Review Process File · EMBO Molecular Medicine]

RIPK1-mediated immunogenic cell death promotes anti-tumour immunity against soft-tissue sarcoma

H.G. Smith, K. Jamal, J.H.S. Dayal, T. Tenev, J. Kyula-Currie, N. Guppy, P. Gazinska, V. Roulstone, G. Liccardi, E. Davies, I. Roxanis, A. Melcher, A.J. Hayes, G.J. Inman, K.J. Harrington, and P. Meier

Review timeline:	Submission date:	11th Jun 2019
	Editorial Decision:	26th Jun 2019
	Revision received:	18th Feb 2020
	Editorial correspondence:	12th Mar 2020
	Authors' correspondence:	18th Mar 2020
	Editorial Decision:	22nd Mar 2020
	Revision received:	15th Apr 2020
	Accepted:	20th Apr 2020

Editor: Lise Roth

Transaction Report:

1st Editorial Decision

26th Jun 2019

Thank you for the submission of your manuscript to EMBO Molecular Medicine. We have now heard back from the two referees who were asked to evaluate your manuscript.

As you will see from the reports below, while they mention the novelty and potential translational interest of the study, they also raise substantial concerns on your work, regarding both the substance (immunogenic tumor cell death should be further demonstrated and additional *in vivo* model(s) should be used) and the form (lack of clarity in the text, missing method description, overstatements regarding the conclusions), which should be addressed in a major revision of the present manuscript.

As mentioned above, it will be important to convincingly show that incorporation of a Smac mimetic in the ILP treatment triggers immunogenic tumour cell death and long-term protection against relapse, and to increase the translational impact with additional *in vivo* model(s). Moreover, addressing the other reviewers' concerns will be necessary for further considering the manuscript in our journal. As revising the manuscript according to the referees' recommendations appears to require a lot of additional work and experimentation, and given the potential interest of your findings, we are ready to extend the deadline to 6 months with the understanding that acceptance of the manuscript would entail a second round of review.

***** Reviewer's comments *****

Referee #1 (Remarks for Author):

The study assesses the ability of cIAP inhibition and immune checkpoint inhibitors to improve the efficacy of TNF plus melphalan therapy in isolated limb perfusion treatment of non-resectable advanced extremity tumours. The study assessed whether cIAP inhibition along TNF/Mel ILP enhances anti-tumour immune reaction and prolong survival and whether addition of immune

checkpoint inhibitors further extends survival in a rat disease model.

The study has a broad scope ranging from in vitro assessment of complex II formation by TNF to in vivo immune infiltration and immune cell activation to long-term anti-tumour immune reactivity. I have two major concerns with the study.

Comment 1: The first is that possibly due to the very broad scope of the study, the in vivo experiments are not sufficiently in depth to fully support the conclusions the authors made. More specifically, the increased immune infiltration is solely based on a flow cytometric measurement of dissociated resected tumours and immune cell activation on the measurement of a single component (granzyme B).

For immune infiltration, the percentage of immune cells within the live cell population is shown. Since this is a relative measure, taking into consideration that the TNF/Mel/SM treatment induces higher level of tumour cell death than TNF/Mel, it may purely be a result of altered proportions of the resident cellular composition of the tumour rather than newly arriving-, i.e. treatment-induced immune cell infiltration.

Immunohistochemical detection of the different immune cell populations or equivalent detection should be included. These studies would also show the localisation of the immune cells within the tumour (whether they are only on the edge or deeply infiltrated in the tumour tissue).

Similarly, the conclusion that inhibition of cIAP leads to enhanced effector immune cell activity in the tumour would need to be corroborated by measurement of a broader range of activation markers in addition to granzyme B protein expression. The authors could consider CD107a surface expression, expression of activating receptors on NK cells, or expression of IFN γ .

Comment 2: The authors re-implanted resected tumours in order to study long-term immune tumour activity. My question regarding this method is how did the authors normalise the amount of tumour cells re-implanted across the different treatment groups. The resected tumour piece from the more effective, TNF/Mel/SM treatment arm, where the tumour size was much smaller on the day of resection (day 10) and very likely even the same sized tumour tissue contained less number of tumour cells. Implantation of this tissue would have much lower number of tumour cells, naturally requiring longer time to reach logarithmic growth. Equally importantly, if the immune system was primed and active against the tumour cells, no tumour growth should have taken place.

Comment 3: The manuscript would require substantial work to make sure all information is entirely correct and essential details are included. I listed a number of examples under 2 minor comments.

Minor comments:

Comment 1: The abstract is somewhat misleading by stating that the authors "report the first clinical application", the work provides only pre-clinical results as a foundation for the clinical testing of a treatment combination.

Comment 2: The first paragraph of the Introduction lacks references.

Comment 3: Introduction, paragraph 2: The following sentence is an overstatement: "Importantly, RIPK1-mediated cell death also operates as the dominant immune surveillance strategy by cytotoxic lymphocytes (Kearney et al., 2018)." The referenced article does not provide evidence that RIPK1-mediated cell death is the dominant immune surveillance strategy in the body (RIPK1 is only mentioned once in the article, in the discussion).

The same is true to the next sentence ("Accordingly, tumour immune evasion arises through loss of TNF..."). In fact, the cited article shows that TNF and IFN signalling are amongst the dominant pathways altered in the ovalbumin-expressing MC38 cell line-based immune evasion model used. The article does not provide evidence that the TNF pathway is the sole, predominant immune evasion pathway of all tumours, only in the one model cell line used in the study.

Please also cite the article introducing the term, immunogenic cell death.

Comment 4: Results, page 7, paragraph 1: incorrect wording: stand-of-care treatment protocol - I believe the authors mean standard of care protocol. The statistics provided in this paragraph are very broad and not supported by references. Please specify the provided 60% value of response to treatment; was it complete response or partial response and what kind of extremity tumour(s)? Also, does the 12-month to disease progression data refers to the medial progression free survival and of which type of extremity malignancy?

Comment 5: In vitro studies with BN175 cells: I assume the dosage of melphalan, smac mimetic and TNF was determined in pilot experiments. Please include this data with a brief justification for the choice of doses used. Most importantly, how does the chosen dose of melphalan correspond to the clinically-relevant dose?

Comment 6: Mechanism of cell death and Complex II formation: The results show that TNF/Mel

treatment is slower to induce cell death and at the early timepoint of 8 h, there is no caspase-8 activation, DEVDase activity or complex II formation. What is the mechanism of cell death induced by TNF/Mel? The lack of DEVDase activity contradicts the cited fact that Mel kills cell via the intrinsic apoptotic pathway. The authors also conclude that Mel triggers outer mitochondrial membrane permeabilisation, but there is no data presented to support this statement.

Comment 7: Supplementary Figure 1B: Is RIPK1i a pharmacological RIPK1 inhibitor or RIPK1 knockout? If inhibitor, please specify in the figure legend the inhibitor used, its concentration and how it was added (pre-treatment or simultaneous).

Comment 8: DEVDase assay: How was the DEVDase activity calculated and what is the unit of the activity? The methods section states that the samples were incubated for up to 24 h for measuring DEVDase activity. This is very unusual and very imprecise description of the method. Why did the samples require such a long incubation time? Also, why were they incubated at room temperature rather than at 37°C? For such long incubation times, non-specific proteolytic activation of caspases in some samples may have taken place?

Comment 9: The statement on ICOS expression of the Treg population has to be rephrased: "Although not statistically significant, TNF/Mel/SM also led to increased expression of the activation marker ICOS on T-reg cells, when compared to TNF/Mel (Fig 4I)."

Referee #2 (Remarks for Author):

Sarcomas are often diagnosed at an advanced stage where metastatic spread occurred already. Sarcomas account for >20% of all pediatric solid malignant cancers and less than 1% of all adult solid malignant cancers. Soft tissue sarcomas are the most frequent, fibrosarcomas account for roughly 7% of all sarcomas. Such numbers should be introduced to the reader and the sarcoma type to be studied should be extended as overall suggestion to the study authors. This written the single fibrosarcoma translational model has value, but it is too limited as a single model, which is the real weakness of the study. The title "RIPK1-mediated immunogenic cell death promotes anti-tumour immunity against soft-tissue sarcoma" by Smith and colleagues is kept very broad. One needs to question if the data support the main conclusion to move to a clinical trial, hope in patients should only be raised with careful pre-clinical model work and data facts. Overall, the study uses a combination of drugs that trigger either the intrinsic or extrinsic apoptotic cancer cell machinery. The intrinsic death pathway is also called the mitochondrial pathway and characterized by permeabilization of the mitochondria induced by a variety of stress signals such as chemotherapeutic drugs or radiation. The authors make use of that and also of the extrinsic death machinery. This one is characterized by triggering of death ligands such as Fas/Apo-1, tumor necrosis factor α , Apo2L/TRAIL to their corresponding cognate death receptors, which are usually CD95/FasR, TNFR1 and DR4/DR5 on the cancer cell-surface. Therefore, the study has merit and uses combinatorial treatment for aggressive sarcomas, but the animal model system is too limited as major critics.

The Figures and the text are well performed and the study concept is all well designed. Overall, the study was a careful one, which could have an impact on better therapeutic intervention on let's say fibrosarcoma treatment, but the broad conclusion must be called an overstatement. The authors have not convincingly shown that they have a new better treatment for soft tissue sarcomas. Thus, the study runs to short from animal model analysis with a single fibrosarcoma rat model system that was applied.

The treatment used is sophisticated and complicated from animal model handling, but the title as well as the papers main conclusion as stated at least three times in the paper suggest that the findings warrant a clinical trial for soft tissue sarcomas with perfusion of drugs limited to limbs e.g. However, as a reviewer one has to critically judge the model system in analysis and more in depth analysis pulling in also PDX models of human origin that also work in immunosuppressed rats could be conducted. The reviewer understands that the immunoregulation is an important point of the study, but then the authors could have used their syngeneic rat model at least also in immunosuppressed rats to demonstrate the mode of action with immune cells better; the impact should be not there in immunodeficient rodents. Thus, the overall critics are that the statement for clinical trial initiation based on the findings requires more model work. This written a genetic sarcoma model would be beneficial; not relying on transplant models of less characterized or standardized sarcoma cell lines that should be properly authenticated.

Major criticisms is given as follows:

1) According to the pre-clinical data shown the study relies heavily on one cell line model, certainly a genetic rodent sarcoma model would be superior. The authors switch often the cancer cell type, one human fibrosarcoma and one rat fibrosarcoma cell line were used as well as three melanoma cell lines and another human liposarcoma cell line. The reviewer had to look that information up and the result, nor the Figure, nor the Figure legend were clear on the origin and cancer type of the cell lines used. This has to be improved and clearly described. A good Figure has that written e.g. on top of the title if distinct cancer types are used for different Figures like in this study. Melanoma cell lines might be for the study irrelevant and too diverse. Sarcomas have a low mutational frequency and thus low neo-antigens as would e.g. all childhood sarcomas have, but melanoma is one of the most mutated cancer types and it displays many neo-antigens. Thus, checkpoint inhibitors will work with their therapy quite different in sarcomas or melanomas. One should separate these experiments from each other and focus on sarcomas only. One should extend on one sarcoma type, where fibrosarcoma could be the focus due to the rat model used. That is a limitation as the authors also write in their discussion, only a single sarcoma model has to be called limited. As stated in the overall summary for study judgment above more could have been done (e.g. with or without immune cell depletion), despite that the reviewer acknowledges the pre-clinical value of the model used, for a journal like EMBO Mol. Med. at least two distinct models on fibrosarcoma that control each other would be essential to strengthen the conclusions. A second model or a paired cell line genetic system lacks for the study, which could have also increased the relevance for pre-clinical data presentation. Certainly, better would be at least two distinct genetic fibrosarcoma models, the species origin and test system for a specific sarcoma type is less relevant, but due to handling the rat might be the best species. Immunosuppressed rat model with human fibrosarcoma cell line could be one way to go to show a second relevant model, where the therapy success should be ablated due to absence of immune cells. Alternatively, other rat fibrosarcoma models might exist.

2) The authors should describe better ongoing or completed clinical trials with their main therapeutic strategy on SMAC mimetics. Several small-molecule SMAC mimetics are now in clinical trials for cancer treatment.

<https://clinicaltrials.gov/ct2/results?cond=&term=smac+mimetics&cntry=&state=&city=&dist=>

8 studies are listed currently, one is completed with results, one terminated, one withdrawn, some still recruiting patients, etc. Half of the trials are focused on hematopoietic cancers such as MPN, lymphoma or multiple myeloma, other studies focus on solid cancers, if studies are ongoing or were discontinued and major insights from these studies exist than it should be disclosed. The paper discussion or introduction could better emphasize on it. There are also different SMAC mimetics and the authors leave that too much in the dark where they switched to more relevant drug at later point in their study. A better description would be helpful what SMAC mimetic drug might be useful as a therapeutic intervention in soft tissue sarcomas. This written the sarcoma types might respond very differently to the treatment, cancer drivers and genetic makeup of sarcoma types is distinct. Thus, if a broad statement wants to be kept, then a more carefully exploration of distinct sarcoma types would be required to be tested, before a statement like at the end of abstract of introduction, abstract or as given in the discussion is justified to propose SMAC mimetics are a potential therapeutic avenue for often metastasizing sarcomas.

3) The authors write that pharmacologic inhibition of RIPK1 or CASP-8 was conducted, but the data are nowhere to be found (the authors refer to Fig. 1E and Appendix Fig. S1 in the text body). Certainly, pharmacologic intervention with a RIPK1 kinase inhibitor would be superior to be shown in the study. Such data could have been included, the reviewer would call them even superior than the shown genetic interference with knockdown by siRNA (which the authors refer to as "knockout" which is not true), due to translational aspects. It should then also be clarified if RIPK1 kinase inhibitors were used successfully in the clinics or not, the reader should not dig for information on that. Similarly, are CASP-8 inhibitors existent and as such a success or not in clinical routine should be clarified.

4) The study introduction is not condensed enough and Merkel cell carcinoma, malignant melanoma, bone sarcomas, soft tissue sarcomas are quite distinct cancers, a better rational why comparing them is lacking. The reviewer understands that limited current treatment options of limb

perfusion are the standard of care procedure, but that might then be more suited for a clinical special journal. The end of the introduction should contain a summary of the study, which is lacking.

Minor critics:

5) Figure 4B should subclassify better the T-cell subtypes at least for Treg, CTL, or other major CD4 cells.

6) The authors claim that the epitope of the murine CTLA4 antibody is identical since it also recognizes the rat CTLA4 molecule. A simple homology blast should be included in the Supplementary data description with clear marking of the epitope recognition site of the antibody for the murine and rat sequences. If only the peptide region used to raise the antibody can be provided then that is the way to disclose the information. Usually antibody companies give out this information when specifically contacted if not given in the datasheet, which again is not the job of the reader or reviewer to control that.

In summary, the authors conclude that a clinical trial should be initiated on soft tissue sarcomas, but more model work and a focus on fibrosarcoma as a tumor entity could be superior to allow for better judgement.

1st Revision - authors' response

18th Feb 2020

Please see next page.

Response to the reviewers: manuscript EMM-2019-10979

We would like to thank you and the referees for the constructive and helpful comments on our manuscript. We have responded to them in full with new experiments to support the questions raised by the referees. We believe that the reviewer's comments helped to strengthen our manuscript further.

There are now **14 new panels**, which support and significantly expand our previous observations.

Key experiments:

1. To corroborate the generality of our observation, we evaluated the sensitivity to TNF/SM of **6 primary patient derived cutaneous squamous carcinoma lines** [1] (comprising 20% of all non-melanoma skin cancers and that can also be treated with ILP-TNF/Mel), and **7 additional cancer cell lines** from epithelioid Sarcomas and synovial sarcomas. Our new data demonstrates that all 13 lines are sensitive to TNF/SM-induced cell death (**new Fig. 2J-N and new Appendix Fig. S2 I-N**). Taken together, our data suggest that the addition of SM might increase the effectiveness of the current standard-of-care treatment of extremity malignancies.
2. We now demonstrate that Mel induces cell death via the intrinsic (mitochondrial) pathway, whereas TNF/SM promotes caspase-8-mediated (extrinsic) cell death (**new Fig S1D,E**). Thus, the combination of TNF/Mel/SM activates both apoptotic pathways, thereby potentiating the sensitivity of sarcoma and melanoma cells to death. What is more, this treatment combination drives activation of non-canonical NF- κ B, which produces danger signals and alarmin [2].
3. We have followed the reviewer's suggestion and have made every effort to assess the immunogenic mode of cell death promoted by TNF/Mel/SM using immuno-compromised animals (**Rebuttal Fig. 1 and 2**). In particular, we tested two immune-compromised rat models, Fisher rats and Hooded rats. We found that the **vascular anatomy** in the legs of the Fisher and Hooded rats are unfortunately **not suitable** for ILP micro-surgery. This is because these rat strains lack collateral anastomosing circulation that is necessary to sustain the viability of the limb (see detailed description, below). Thus, unlike in the Brown Norway rat strain, the ILP procedure in Fisher and Hooded rats will result in the loss of the entire lower limb distal to the vascular tourniquet. Therefore, we were not able to use these immune-compromised rat strains to address the contribution of the immune system. Unfortunately, immune-deficient Brown Norway rats do not exist, and hence rat-based PDX studies are not possible (**Rebuttal Fig. 2A-H**). Consequently, the BN175 ILP *in vivo* model is the only one available to date.

Since sarcomas are a heterogeneous group of tumours with over 80 subtypes, we believe that testing whether the effectiveness of this novel treatment is affected by biological differences between histological subtypes would be best determined in the context of a clinical trial. The value of this BN175 ILP *in vivo* model is demonstrated by the ongoing **TITAN clinical trial** at the Royal Marsden Hospital <https://clinicaltrials.gov/ct2/show/study/NCT03555032>. **That study has already recruited >10 patients**. All pre-clinical data that formed the basis for the approval of the TITAN trial by the UK Medicines and Healthcare products Regulatory Agency (MHRA) and the Research Ethics Committee came from isolated limb perfusion experiments with TNF/Mel/Oncolytic virus to treat extremity soft tissue sarcoma in Brown Norway rats [3].

Together, our data support the notion that SM augment the efficacy of the standard-of-care treatment for extremity malignancies by promoting RIPK1-dependent cell death.

Please find below a point-by-point response to the reviewers' comments, with the **reviewers' comments in blue boxes** and our response in 'plain text'. Modifications of the main manuscript are highlighted in **yellow (see article file)**.

For convenience, we have included the relevant new data for each reviewer.

Reviewer #1 (Reviewer Comments to the Author):

The study assesses the ability of cIAP inhibition and immune checkpoint inhibitors to improve the efficacy of TNF plus melphalan therapy in isolated limb perfusion treatment of non-resectable advanced extremity tumours. The study assessed whether cIAP inhibition along TNF/Mel ILP enhances anti-tumour immune reaction and prolong survival and whether addition of immune checkpoint inhibitors further extends survival in a rat disease model.

The study has a broad scope ranging from in vitro assessment of complex II formation by TNF to in vivo immune infiltration and immune cell activation to long-term anti-tumour immune reactivity. I have two major concerns with the study.

Comments

1. The first is that possibly due to the very broad scope of the study, the in vivo experiments are not sufficiently in depth to fully support the conclusions the authors made. More specifically, the increased immune infiltration is solely based on a flow cytometric measurement of dissociated resected tumours and immune cell activation on the measurement of a single component (granzyme B).

For immune infiltration, the percentage of immune cells within the live cell population is shown. Since this is a relative measure, taking into consideration that the TNF/Mel/SM treatment induces higher level of tumour cell death than TNF/Mel, it may purely be a result of altered proportions of the resident cellular composition of the tumour rather than newly arriving-, i.e. treatment-induced immune cell infiltration.

Immunohistochemical detection of the different immune cell populations or equivalent detection should be included. These studies would also show the localisation of the immune cells within the tumour (whether they are only on the edge or deeply infiltrated in the tumour tissue).

Similarly, the conclusion that inhibition of cIAP leads to enhanced effector immune cell activity in the tumour would need to be corroborated by measurement of a broader range of activation markers in addition to granzyme B protein expression. The authors could consider CD107a surface expression, expression of activating receptors on NK cells, or expression of IFN γ .

We welcome the reviewer's comment, and have carried out Immuno-histochemical (IHC) analysis to evaluate the presence of different immune populations. Specifically, we used anti-NKR-P1 and anti-FOXP3 antibodies to evaluate the localization and infiltration of rat NK cells and T-regs, respectively.

Since rat-specific antibodies are not available, we first tested whether mouse-specific anti-NKR-P1 and anti-FOXP3 antibodies might be suitable to detect rat-NKR-P1 and rat-FOXP3, respectively. To optimize staining conditions we used rat spleens. Despite numerous attempts and the use of different staining protocols, we were unable to obtain a specific staining pattern for either antibodies in the spleens of rats (**see Rebuttal Figure 1**). The anti-NKR-P1 antibody should show membrane staining of NK cells, which would ordinarily be clustered around the rim of the splenic white pulp follicles. However, we did not detect any such staining. In contrast, we detected some cytoplasmic staining of groups of plasmacytoid cells in the red pulp, which reflects non-specific staining. We also noted cells that stained somewhat darker, but the staining appears to be cytoplasmic and fewer in numbers than expected for NK cells in the spleen. We conclude, therefore, that **the available mouse-specific anti-NKR-P1 antibody is not suitable for IHC in rats**. Therefore, we were unable to identify NK cells with sufficient confidence.

With regards to the anti-FOXP3 antibody, we also experienced specificity issues in rats. While this antibody detects rat T-regs, it also cross-reacts with plasma cells within the trabeculae and surrounding red pulp areas at the periphery of the white pulp follicles in rat spleens. Because of this and due to sensitivity issues, our pathologists could not ascertain with confidence whether FOXP3-positive cells were more abundant upon ILP-TNF/Mel/SM treatment, and whether this cell population decreases following treatment with anti-CTLA-4 antibodies.

Rebuttal Figure R1: Evaluation of the specificity of the anti-NKR-P1 and anti-FOXP3 antibodies in rats. A) The mouse-specific anti-NKR-P1 antibody non-specifically stains plasma cells in the rat spleen. B) The anti-FOXP3 antibody detects both T-regs as well as plasma cells in the spleen, and hence is not specific for rat-T-regs.

2. The authors re-implanted resected tumours in order to study long-term immune tumour activity. My question regarding this method is how did the authors normalise the amount of tumour cells re-implanted across the different treatment groups. The resected tumour piece from the more effective, TNF/Mel/SM treatment arm, where the tumour size was much smaller on the day of resection (day 10) and very likely even the same sized tumour tissue contained less number of tumour cells. Implantation of this tissue would have much lower number of tumour cells, naturally requiring longer time to reach logarithmic growth. Equally importantly, if the immune system was primed and active against the tumour cells, no tumour growth should have taken place.

We apologise for not being clearer about the methodology for the re-challenge experiment. We re-challenged these animals with fresh 1×10^7 cells/ml of **parental** BN175 sarcoma cells. We did NOT re-implant resected tumour chunks. Therefore, the injected cell number was consistent throughout.

Regarding the second point of the reviewer, we agree that the re-challenge experiments showed partial, rather than complete, protection against BN175 tumour cells. However, it should be noted that the BN175 model is a highly aggressive model with BN175 cells proliferating at a very high rate leading to animals being sacrificed at 12 days in all previous control experiments. The delay in growth demonstrated in Figure 6 represents a meaningful alteration in tumour growth kinetics in this highly proliferative model system.

3. The manuscript would require substantial work to make sure all information is entirely correct and essential details are included. I listed a number of examples under 2 minor comments.

Minor comments:

Comment 1: The abstract is somewhat misleading by stating that the authors "report the first clinical application", the work provides only pre-clinical results as a foundation for the clinical testing of a treatment combination.

We apologise for not including all information. This has now been corrected.

We also have improved the abstract.

2. The first paragraph of the Introduction lacks references.

We have corrected our manuscript to include the missing references.

3. Introduction, paragraph 2: The following sentence is an overstatement: "Importantly, RIPK1-mediated cell death also operates as the dominant immune surveillance strategy by cytotoxic lymphocytes (Kearney et al., 2018)." The referenced article does not provide evidence that RIPK1-mediated cell death is the dominant immune surveillance strategy in the body (RIPK1 is only mentioned once in the article, in the discussion). The same is true to the next sentence ("Accordingly, tumour immune evasion arises through loss of TNF..."). In fact, the cited article shows that TNF and IFN signalling are amongst the dominant pathways altered in the ovalbumin-expressing MC38 cell line-based immune evasion model used. The article does not provide evidence that the TNF pathway is the sole, predominant immune evasion pathway of all tumours, only in the one model cell line used in the study. Please also cite the article introducing the term, immunogenic cell death.

We apologise for these oversights and have amended our manuscript accordingly.

4. Results, page 7, paragraph 1: incorrect wording: stand-of-care treatment protocol - I believe the authors mean standard of care protocol. The statistics provided in this paragraph are very broad and not supported by references. Please specify the provided 60% value of response to treatment; was it complete response or partial response and what kind of extremity tumour(s)? Also, does the 12-month to disease progression data refers to the medial progression free survival and of which type of extremity malignancy?

We have corrected our manuscript as suggested.

The 60% response rate to ILP-TNF/Mel treatment refers to a palliative protocol for extremity soft tissue sarcoma [4]. More accurately, the overall response rate ranges from 53-87% (complete 16-56%, partial 31-68%) allowing amputation to be avoided in 62-94% of patients.

The '12-month to disease progression data' refers to the median progression-free survival for extremity soft tissue sarcoma following ILP-TNF/Mel in the palliative protocol [5]. This has now been clarified and cross-referenced in the manuscript.

5. In vitro studies with BN175 cells: I assume the dosage of melphalan, smac mimetic and TNF was determined in pilot experiments. Please include this data with a brief justification for the choice of doses used. Most importantly, how does the chosen dose of melphalan corresponds to the clinically-relevant dose?

Yes indeed, the dosing of these compounds was determined in pilot experiments. These pilot experiments and the evaluation of the right dosage for melphalan and TNF in rats were previously published [3, 6, 7]. The 100 µg dose of Melphalan equates to a clinically relevant dose of 50 mg. The dosage for SMAC mimetics, and in particular Birinapant, has previously been established for mouse as 10 mg/kg [8]. By applying the FDA guidelines for dose conversion between species, this equates to 5 mg/kg for rats [9] which is what we used in our *in vivo* ILP model.

→ We have added this to the Method section for clarification.

6. Mechanism of cell death and Complex II formation: The results show that TNF/Mel treatment is slower to induce cell death and at the early timepoint of 8 h, there is no caspase-8 activation, DEVDase activity or complex II formation. What is the mechanism of cell death induced by TNF/Mel? The lack of DEVDase activity contradicts the cited fact that Mel kills cell via the intrinsic apoptotic pathway. The authors also conclude that Mel triggers outer mitochondrial membrane permeabilisation, but there is no data presented to support this statement.

The kinetics of Mel killing is such that DEVDase activity can only be detected at later time points. At 8 hrs (an early time point) cell death is not yet detectable and hence no caspase activity can be measured. The 8 hrs time point was included because of the TNF/SM treatment, which activates the extrinsic death receptor pathway early on. We have now included a 24 hrs time point (**Appendix. Fig. S1D**, see below), which clearly shows that Mel and TNF/Mel induce caspase activation, and that this is rescued by treatment with the caspase inhibitor zVAD-FMK.

The mechanism of Melphalan-induced cell death was previously established [10-12]. These reports demonstrate that treatment with Mel kills cells via intrinsic apoptosis. We have corroborated these findings in our revised manuscript. In particular, we generated HT1080 cells that express BCL2, which blocks mitochondrial cell death [13]. While treatment with Mel and TNF/Mel significantly sensitised control HT1080 cells, the same treatments had little effect in BCL2-expressing HT1080 cells (**Appendix. Fig. S1E**, see below). This corroborates previous reports indicating that Mel induces intrinsic apoptosis [10-12].

BN175 ■ Standard-of-care (TNF/Mel)

Appendix. Fig. S1D

Appendix. Fig. S1E

7. Supplementary Figure 1B: Is RIPK1i a pharmacological RIPK1 inhibitor or RIPK1 knockout? If inhibitor, please specify in the figure legend the inhibitor used, its concentration and how it was added (pre-treatment or simultaneous).

RIPK1i refers to the pharmacological inhibitor of RIPK1 (GSK'963). We have corrected our manuscript to clearly state what RIPK1i stands for.

100 nM of RIPK1i (GSK'963) was added together with the other drugs. The figure legends were corrected accordingly.

8. DEVDase assay: How was the DEVDase activity calculated and what is the unit of the activity? The methods section states that the samples were incubated for up to 24 h for measuring DEVDase activity. This is very unusual and very imprecise description of the method. Why did the samples require such a long incubation time? Also, why were they incubated at room temperature rather than at 37°C? For such long incubation times, non-specific proteolytic activation of caspases in some samples may have taken place?

The DEVDase assay was performed as previously described [13-16]. We have modified the Method section to provide more details.

9. The statement on ICOS expression of the Treg population has to be rephrased:

"Although not statistically significant, TNF/Mel/SM also led to increased expression of the activation marker ICOS on T-reg cells, when compared to TNF/Mel (Fig 4I)."

This was corrected.

Reviewer #2 (Reviewer Comments to the Author):

Sarcomas are often diagnosed at an advanced stage where metastatic spread occurred already. Sarcomas account for >20% of all pediatric solid malignant cancers and less than 1% of all adult solid malignant cancers. Soft tissue sarcomas are the most frequent, fibrosarcomas account for roughly 7% of all sarcomas. Such numbers should be introduced to the reader and the sarcoma type to be studied should be extended as overall suggestion to the study authors.

These numbers are now included (see Introduction).

This written the single fibrosarcoma translational model has value, but it is too limited as a single model, which is the real weakness of the study. The title "RIPK1-mediated immunogenic cell death promotes anti-tumour immunity against soft-tissue sarcoma" by Smith and colleagues is kept very broad. One needs to question if the data support the main conclusion to move to a clinical trial, hope in patients should only be raised with careful pre-clinical model work and data facts. Overall, the study uses a combination of drugs that trigger either the intrinsic or extrinsic apoptotic cancer cell machinery. The intrinsic death pathway is also called the mitochondrial pathway and characterized by permeabilization of the mitochondria induced by a variety of stress signals such as chemotherapeutic drugs or radiation. The authors make use of that and also of the extrinsic death machinery. This one is characterized by triggering of death ligands such as Fas/Apo-1, tumor necrosis factor α , Apo2L/TRAIL to their corresponding cognate death receptors which are usually CD95/FasR, TNFR1 and DR4/DR5 on the cancer cell-surface. Therefore, the study has merit and uses combinatorial treatment for aggressive sarcomas, but the animal model system is too limited as major critics.

The Figures and the text are well performed and the study concept is all well designed. Overall, the study was a careful one, which could have an impact on better therapeutic intervention on let's say fibrosarcoma treatment, but the broad conclusion must be called an overstatement. The authors have not convincingly shown that they have a new better treatment for soft tissue sarcomas. Thus, the study runs to short from animal model analysis with a single fibrosarcoma rat model system that was applied.

The treatment used is sophisticated and complicated from animal model handling, but the title as well as the papers main conclusion as stated at least three times in the paper suggest that the findings warrant a clinical trial for soft tissue sarcomas with perfusion of drugs limited to limbs e.g. However, as a reviewer one has to critically judge the model system in analysis and more in depth analysis pulling in also PDX models of human origin that also work in immunosuppressed rats could be conducted. The reviewer understands that the immunoregulation is an important point of the study, but then the authors could have used their syngeneic rat model at least also in immunosuppressed rats to demonstrate the mode of action with immune cells better; the impact should be not there in immunodeficient rodents. Thus, the overall critics are that the statement for clinical trial initiation based on the findings requires more model work. This written a genetic sarcoma model would be beneficial; not relying on transplant models of less characterized or standardized sarcoma cell lines that should be properly authenticated.

We appreciate the thoughtful response by the reviewer. We acknowledge the limitation of having only one *in vivo* model for ILP. While this model is the only ILP model available world-wide, it has provided sufficient pre-clinical rationale for the initiation of a clinical trial (having been reviewed in the data package submitted to the MHRA and Research Ethics Committees). Accordingly, the pre-clinical benefit of isolated limb perfusion with TNF/Mel/Oncolytic virus in treating extremity soft tissue sarcoma in Brown Norway rats [3] formed the basis for the **TITAN clinical trial** at the Royal Marsden Hospital <https://clinicaltrials.gov/ct2/show/study/NCT03555032>. Importantly, the ongoing TITAN phase I/II clinical trial has recruited >10 patients with extremity malignancies such as metastatic melanoma, epithelioid sarcoma, synovial sarcoma, fibrosarcoma, liposarcoma and cutaneous squamous cell carcinoma.

To date, the syngeneic BN175 ILP rat model is the only *in vivo* model of its kind. Unfortunately, the isolated limb perfusion procedure is not suitable for studies on mice because the blood vessels are too small for femoral artery and vein cannulation, which is needed to provide a closed circuit for TNF/Mel perfusion. Thus, we are restricted to the use of rats to carry out our ILP studies. The syngeneic BN175 rat

line is a high-grade pleomorphic sarcoma, thus these cells are not strictly a fibrosarcoma type. Since sarcomas are a heterogeneous group of tumours with over 80 subtypes, we believe that testing whether the effectiveness of this novel treatment is affected by biological differences between histological subtypes would be best determined in the context of a clinical trial.

Nevertheless, in an attempt to expand the generality of our findings, we evaluated the sensitivity of a larger panel of primary STS cell lines [4]. Our data demonstrate that these different STS lines are all sensitive to the addition of SM.

- **Epithelioid Sarcoma** comprise less than 1% of STS cases.
 - o HS-ES-2M
 - o HS-ES-2R
 - o HS-ES-1
- **Synovial Sarcoma** comprise 10% of STS cases.
 - o SW982
 - o HS-SY-II

Epithelioid Sarcoma

Figure 2J and Appendix Figure S2 I-J

Synovial sarcoma

Figure 2K and Appendix Figure S2K

In addition, isolated limb perfusion with TNF/Mel has been used to treat cutaneous squamous cell carcinomas (cSCC) that are not suitable for surgical resection. In fact, usually cSCC present as local lesions and thus can be treated with the ILP method with overall response rates of 81% [17]. Accordingly, we evaluated the efficacy of TNF/SM in panel 6 cSCC lines, five of which were recently patient-derived [1]. Importantly, MET1, MET2, MET4, IC8 and T1 cSCC cell lines have all been derived from local limb occurrences.

cSCC

Figure 2L-N and Appendix Figure S2L-N

cSCC-MET1, cSCC-MET2 and cSCC-MET4 were isolated from the same patient from the primary, recurrence and metastatic tumour sites, respectively [18]. cSCCIC8 was isolated from an immune competent patient and cSCC-T1 from a transplant patient who was a transplant recipient.

Taken together, our data suggest that the addition of SM might increase the effectiveness of the current standard-of-care treatment of a broad group of extremity malignancies.

Major critics is given as follows:

1) According to the pre-clinical data shown the study relies heavily on one cell line model, certainly a genetic rodent sarcoma model would be superior. The authors switch often the cancer cell type, one human fibrosarcoma and one rat fibrosarcoma cell line were used as well as three melanoma cell lines and another human liposarcoma cell line. The reviewer had to look that information up and the result, nor the Figure, nor the Figure legend were clear on the origin and cancer type of the cell lines used. This has to be improved and clearly described. A good Figure has that written e.g. on top of the title if distinct cancer types are used for different Figures like in this study. Melanoma cell lines might be for the study irrelevant and too diverse. Sarcomas have a low mutational frequency and thus low neo-antigens as would e.g. all childhood sarcomas have, but melanoma is one of the most mutated cancer types and it displays many neo-antigens. Thus, checkpoint inhibitors will work with their therapy quite different in sarcomas or melanomas. One should separate these experiments from each other and focus on sarcomas only. One should extend on one sarcoma type, where fibrosarcoma could be the focus due to the rat model used. That is a limitation as the authors also write in their discussion, only a single sarcoma model has to be called limited. As stated in the overall summary for study judgment above more could have been done (e.g. with or without immune cell depletion), despite that the reviewer acknowledges the pre-clinical value of the model used, for a journal like EMBO Mol. Med. at least two distinct models on

fibrosarcoma that control each other would be essential to strengthen the conclusions. A second model or a paired cell line genetic system lacks for the study, which could have also increased the relevance for pre-clinical data presentation. Certainly, better would be at least two distinct genetic fibrosarcoma models, the species origin and test system for a specific sarcoma type is less relevant, but due to handling the rat might be the best species. Immunosuppressed rat model with human fibrosarcoma cell line could be one way to go to show a second relevant model, where the therapy success should be ablated due to absence of immune cells. Alternatively, other rat fibrosarcoma models might exist.

We are unfortunately restricted to the use of Brown Norway rats to carry out ILP studies due to technical reasons. The isolated limb perfusion procedure is not suitable for studies in mice because murine blood vessels are too small for femoral artery and vein cannulation. We have attempted such procedures in the past and have consistently failed to achieve a successful perfusion circuit or to maintain limb viability following vascular access.

To follow the reviewer's suggestions and extend our findings to clinically relevant PDX models or human fibrosarcoma models, we investigated whether we could use immune-compromised animals.

All our studies were conducted in Brown Norway rats. Unfortunately, however, immune-compromised Brown Norway rats are not commercially available.

Therefore, we evaluated the suitability of immune compromised **Fisher rats** and **Hooded pigmented rats**.

In order for the isolated limb perfusion to be achievable, rats need to have a particular anatomical arrangement of the femoral blood vessels. Specifically, rats need to have an extensive collateral circulation via the internal iliac artery to the leg. This allows ligation of the main femoral vessels during ILP without compromising the viability of the limb distal to the surgical tourniquet. It needs to be noted that the ligated/cannulated artery cannot be unligated and remains completely occluded at the end of the surgical procedure. Therefore, the rat needs to have additional arteries/veins that can bypass/restore blood circulation to the leg immediately after release of the tourniquet [19]. While Brown Norway rats have several superficial inferior epigastric arteries, not every strain of rat has a similar extensive arrangement of femoral blood vessels.

Therefore, we assessed the suitability of the ILP procedure in immune-compromised Fisher rats and Hooded pigmented rats. To evaluate collateral circulation after ILP, we used the dye methylene blue, which was added to the perfusion circuit during ILP, staining the leg blue. After release of the tourniquet post-ILP, the blue dye should be washed out quickly. The test monitors collateral circulation, which would allow the drainage of the dye post-ILP. As shown in Rebuttal Fig. 2, the dye is not draining from the hind limb post-ILP in Hooded pigmented rats (**Rebuttal Fig. 2**) and Fisher rats (data not shown). This indicates the lack of collateral circulation in both these rat strains. Unfortunately, this means that the athymic Hooded pigmented rats and Fisher rats cannot be used for the ILP procedure as the hind leg would become necrotic due to the absence of collateral circulation.

Brown Norway rats have a superficial inferior epigastric artery and vein, which arise above the level of the tourniquet and allow collateral circulation following ligation of the femoral vessels (see below for details).

Hooded pigmented rat (Rebuttal Figure 2):

A) Blue dye given via ILP

B) Lack of drainage of Blue dye post ILP

C) Visually there is no Superficial Inferior Epigastric Artery

D)

E)

D,E) Perfusion field delineation by methylene blue infusion in the Fisher 344 (D) and Brown Norway (E) rat.

F)

G)

F,G) Flow of methylene blue through collateral circulation after removal of tourniquet, as indicated by cutaneous distribution on the dorsal (F) and ventral (G) surface.

H) **Brown Norway rat:** Microscopic view (5x magnification) of Brown Norway femoral vessels (A), indicating site of origin of profunda femoris vessels (P) and superficial inferior epigastric vessels (SIE) relative to abdominal wall (dotted line).

⇒ **Conclusion:** The isolated limb perfusion procedure cannot be conducted in Fisher and Hooded pigmented rats as they lack the necessary blood vessel arrangements.

2) The authors should describe better ongoing or completed clinical trials with their main therapeutic strategy on SMAC mimetics. Several small-molecule SMAC mimetics are now in clinical trials for cancer treatment.

<https://clinicaltrials.gov/ct2/results?cond=&term=smac+mimetics&cntry=&state=&city=&dist=>

8 studies are listed currently, one is completed with results, one terminated, one withdrawn, some still recruiting patients, etc. Half of the trials are focused on hematopoietic cancers such as MPN, lymphoma or multiple myeloma, other studies focus on solid cancers, if studies are ongoing or were discontinued and major insights from these studies exist than it should be disclosed. The paper discussion or introduction could better emphasize on it. There are also different SMAC mimetics and the authors leave that too much in the dark where they switched to more relevant drug at later point in their study. A better description would be helpful what SMAC mimetic drug might be useful as a therapeutic intervention in soft tissue sarcomas.

This written the sarcoma types might respond very differently to the treatment, cancer drivers and genetic makeup of sarcoma types is distinct. Thus, if a broad statement wants to be kept, then a more carefully exploration of distinct sarcoma types would be required to be tested, before a statement like at the end of abstract of introduction, abstract or as given in the discussion is justified to propose SMAC mimetics are a potential therapeutic avenue for often metastasizing sarcomas.

We have followed the reviewer's suggestion and have expanded our manuscript (see **highlighted** section in the Introduction) to include a discussion of ongoing and completed clinical trials with SMAC mimetics. We have also highlighted the different types of SMAC mimetics available and which ones might be best to use in a therapeutic intervention setting for soft tissue sarcomas.

Briefly, we have indicated the following:

In summary, SM-164 and Birinapant are bivalent SMAC mimetics, meaning that they have a very similar mechanism of action as they both bind to the BIR2 and BIR3 domains of Inhibitor of apoptosis (IAP) proteins to promote their autoubiquitylation and degradation. In fact, we find that Birinapant and SM-164 display the same efficacy *in vitro*, and kill BN175 cells (see Fig S3A) to the same extent when combined with TNF/Mel.

Many different SMAC mimetics are in clinical trials for a variety of malignancies. Lately, SMAC mimetics are evaluated in combination with immune checkpoint blockade. The use of SMAC mimetic has shown limited clinical efficacy. This is mainly because SMAC mimetic efficacy heavily relies on the ability of cells to produce autocrine TNF [20, 21]. Our study circumvents this problem as TNF is exogenously administered via ILP. At present the cytotoxic potential of TNF is not exploited. This could be changed via the addition of SMAC mimetics to this regimen. The use of SMAC mimetics not only enhances the cytotoxic potential of TNF but SMAC mimetics also increase antigen processing machinery [22].

- We have added further details in the text about the current clinical progress with SMAC mimetics in cancer.

3) The authors write that pharmacologic inhibition of RIPK1 or CASP-8 was conducted, but the data are nowhere to be found (the authors refer to Fig. 1E and Appendix Fig. S1 in the text body). Certainly, pharmacologic intervention with a RIPK1 kinase inhibitor would be superior to be shown in the study. Such data could have been included, the reviewer would call them even superior than the shown genetic interference with knockdown by siRNA (which the authors refer to as "knockout" which is not true), due to translational aspects. It should then also be clarified if RIPK1 kinase inhibitors were used successfully in the clinics or not, the reader should not dig for information on that. Similarly, are CASP-8 inhibitors existent and as such a success or not in clinical routine should be clarified.

We welcome the reviewer's suggestions and apologies for the inconsistencies and oversight. The data with the RIPK1 inhibitor are shown in Appendix Fig S1B.

RIPK1 inhibitors (GSK2982772) are currently in phase II clinical trials for psoriasis, rheumatoid arthritis and ulcerative colitis [23, 24]. The inflammation in this disease setting is caused by excessive cell death, which is targeted by the RIPK1 inhibitor [25, 26].

RIPK1 inhibitor is also in phase I clinical trial for pancreatic cancer and other tumours [27] as inhibiting its kinase activity in tumour-associated macrophages was shown to drive cytotoxic T-cell activation leading to anti-tumour immunity.

There are no selective caspase-8 inhibitors but the pan-caspase inhibitor Emricasan was in clinical trials for nonalcoholic steatohepatitis (NASH)-related cirrhosis with severe portal hypertension (PH), but failed to meet the primary endpoint and so is no longer in clinical development (<https://ilc-congress.eu/press-release/emricasan-fails-to-meet-primary-endpoint-in-encore-ph-study-but-shows-potential-benefits-in-high-risk-individuals/>).

4) The study introduction is not condensed enough and Merkel cell carcinoma, malignant melanoma, bone sarcomas, soft tissue sarcomas are quite distinct cancers, a better rational why comparing them is lacking. The reviewer understands that limited current treatment options of limb perfusion are the standard of care procedure, but that might then be more suited for a clinical special journal. The end of the introduction should contain a summary of the study, which is lacking.

We welcome the reviewer's suggestions and have included a study summary at the end of the introduction.

Minor critics:

5) Figure 4B should subclassify better the T-cell subtypes at least for Treg, CTL, or other major CD4 cells.

We welcome the reviewer's suggestions, however, due to the lack of specific rat antibodies for immune-profiling via FACS such as IFN gamma we are unable to classify CD3⁺CD8⁺ T cells as cytotoxic.

Minor critics:

6) The authors claim that the epitope of the murine CTLA4 antibody is identical since it also recognizes the rat CTLA4 molecule. A simple homology blast should be included in the Supplementary data description with clear marking of the epitope recognitions site of the antibody for the murine and rat sequences. If only the peptide region used to raise the antibody can be provided then that is the way to disclose the information. Usually antibody companies give out this information when specifically contacted if not given in the datasheet, which again is not the job of the reader or reviewer to control that.

In summary, the authors conclude that a clinical trial should be initiated on soft tissue sarcomas, but more model work and a focus on fibrosarcoma as a tumor entity could be superior to allow for better judgement.

We welcome the reviewer's comment. We have tried to obtain the sequence against which the antibody was raised but, unfortunately, the company said the sequence was proprietary information and is not willing to release it. Thus, we are not able to carry out the homology alignment requested (note, murine and rodent CTLA-4 protein share 86% homology).

We have assessed the cross reactivity as illustrated in Appendix Fig S4A and the murine antibody does cross-react with the rodent epitope.

References:

1. Hassan, S., et al., *A Unique Panel of Patient-Derived Cutaneous Squamous Cell Carcinoma Cell Lines Provides a Preclinical Pathway for Therapeutic Testing*. Int J Mol Sci, 2019. **20**(14).
2. Yatim, N., et al., *RIPK1 and NF-kappaB signaling in dying cells determines cross-priming of CD8(+) T cells*. Science, 2015. **350**(6258): p. 328-34.
3. Smith, H.G., et al., *PD-1 blockade following isolated limb perfusion with vaccinia virus prevents local and distant relapse of soft-tissue sarcoma*. Clin Cancer Res, 2019.
4. Smith, H.G. and A.J. Hayes, *The role of regional chemotherapy in the management of extremity soft tissue malignancies*. Eur J Surg Oncol, 2016. **42**(1): p. 7-17.
5. Smith, H.G., et al., *Isolated Limb Perfusion with Melphalan and Tumour Necrosis Factor alpha for In-Transit Melanoma and Soft Tissue Sarcoma*. Ann Surg Oncol, 2015. **22 Suppl 3**: p. S356-61.
6. Pencavel, T.D., et al., *Isolated limb perfusion with melphalan, tumour necrosis factor-alpha and oncolytic vaccinia virus improves tumour targeting and prolongs survival in a rat model of advanced extremity sarcoma*. Int J Cancer, 2015. **136**(4): p. 965-76.
7. Wilkinson, M.J., et al., *Isolated limb perfusion with biochemotherapy and oncolytic virotherapy combines with radiotherapy and surgery to overcome treatment resistance in an animal model of extremity soft tissue sarcoma*. Int J Cancer, 2016. **139**(6): p. 1414-22.
8. Lalaoui, N., et al., *Targeting p38 or MK2 Enhances the Anti-Leukemic Activity of Smac-Mimetics*. Cancer Cell, 2016. **30**(3): p. 499-500.
9. Nair, A.B. and S. Jacob, *A simple practice guide for dose conversion between animals and human*. J Basic Clin Pharm, 2016. **7**(2): p. 27-31.
10. Gomez-Bougie, P., et al., *Melphalan-induced apoptosis in multiple myeloma cells is associated with a cleavage of Mcl-1 and Bim and a decrease in the Mcl-1/Bim complex*. Oncogene, 2005. **24**(54): p. 8076-9.
11. Matsura, T., et al., *Endogenously generated hydrogen peroxide is required for execution of melphalan-induced apoptosis as well as oxidation and externalization of phosphatidylserine*. Chem Res Toxicol, 2004. **17**(5): p. 685-96.
12. Park, G.B., et al., *Melphalan-induced apoptosis of EBV-transformed B cells through upregulation of TAp73 and XAF1 and nuclear import of XPA*. J Immunol, 2013. **191**(12): p. 6281-91.
13. Tenev, T., et al., *The Ripoptosome, a signaling platform that assembles in response to genotoxic stress and loss of IAPs*. Mol Cell, 2011. **43**(3): p. 432-48.
14. Feltham, R., et al., *Mind Bomb Regulates Cell Death during TNF Signaling by Suppressing RIPK1's Cytotoxic Potential*. Cell Rep, 2018. **23**(2): p. 470-484.

15. Jaco, I., et al., *MK2 Phosphorylates RIPK1 to Prevent TNF-Induced Cell Death*. Mol Cell, 2017. **66**(5): p. 698-710 e5.
16. Liccardi, G., et al., *RIPK1 and Caspase-8 Ensure Chromosome Stability Independently of Their Role in Cell Death and Inflammation*. Mol Cell, 2019. **73**(3): p. 413-428 e7.
17. Huis In 't Veld, E.A., et al., *Isolated limb perfusion for unresectable extremity cutaneous squamous cell carcinoma; an effective limb saving strategy*. Br J Cancer, 2018. **119**(4): p. 429-434.
18. Proby, C.M., et al., *Spontaneous keratinocyte cell lines representing early and advanced stages of malignant transformation of the epidermis*. Exp Dermatol, 2000. **9**(2): p. 104-17.
19. Manusama, E.R., et al., *Isolated limb perfusion with TNF alpha and melphalan in a rat osteosarcoma model: a new anti-tumour approach*. Eur J Surg Oncol, 1996. **22**(2): p. 152-7.
20. Fulda, S., *Promises and Challenges of Smac Mimetics as Cancer Therapeutics*. Clin Cancer Res, 2015. **21**(22): p. 5030-6.
21. Vince, J.E., et al., *IAP antagonists target cIAP1 to induce TNFalpha-dependent apoptosis*. Cell, 2007. **131**(4): p. 682-93.
22. Ye, W., et al., *ASTX660, an antagonist of cIAP1/2 and XIAP, increases antigen processing machinery and can enhance radiation-induced immunogenic cell death in preclinical models of head and neck cancer*. Oncoimmunology, 2020. **9**(1): p. 1710398.
23. Harris, P.A., et al., *Discovery of a First-in-Class Receptor Interacting Protein 1 (RIP1) Kinase Specific Clinical Candidate (GSK2982772) for the Treatment of Inflammatory Diseases*. J Med Chem, 2017. **60**(4): p. 1247-1261.
24. Weisel, K., et al., *Randomized clinical study of safety, pharmacokinetics, and pharmacodynamics of RIPK1 inhibitor GSK2982772 in healthy volunteers*. Pharmacol Res Perspect, 2017. **5**(6).
25. Berger, S.B., et al., *Cutting Edge: RIP1 kinase activity is dispensable for normal development but is a key regulator of inflammation in SHARPIN-deficient mice*. J Immunol, 2014. **192**(12): p. 5476-80.
26. Patel, S., et al., *RIP1 inhibition blocks inflammatory diseases but not tumor growth or metastases*. Cell Death Differ, 2019.
27. Wang, W., et al., *RIP1 Kinase Drives Macrophage-Mediated Adaptive Immune Tolerance in Pancreatic Cancer*. Cancer Cell, 2018. **34**(5): p. 757-774 e7.

Thank you for submitting your revised manuscript to EMBO Molecular Medicine. We have now heard back from the two referees who re-evaluated your manuscript.

As you will see from the reports pasted below, while referee #2 is satisfied with the revisions, referee #1 regrets that the issue raised on immune infiltration quantification was not satisfactorily addressed.

I asked this referee for specific recommendations, and he/she stated:

“The major point I raised was not addressed and the conclusions were not changed accordingly. The relative amount on NK cells and T cells increases in the tumour tissue merely due to the clearance of the dead tumour cells. This is not a proof of increased immune infiltration. For the T cell population, the altered frequency of T cell subsets confirm altered immune component in the tumour, but the reduced number of T helper cells and substantially increased number of regulatory T cells argues against immune activation. In order to corroborate the conclusion of the authors that there is increased immune infiltration and NK cell and CD8 CTL activation, additional experiments should have been carried out (either at this stage or for the studies incorporating checkpoint inhibitor targeting (CTLA4, PD1). These could have been:

For corroborating increased infiltration:

1. Determining absolute numbers of NK cells, CD8 CTLs
2. Immunohistochemistry showing infiltration

For corroborating NK and/or CD8 CTL activation:

1. Detection of induction of other activation markers, or repression of checkpoint inhibitors
2. Detection of cytokine secretion (e.g. NK cell isolation followed by IFN γ ELISA)

These studies could have been done either with the ILP-TNF/Mel/SM treatment or for the experiments with inclusion of anti-CTLA-4 or anti-PD-1 antibodies.”

In some cases, we may consult with the authors before deciding on the outcome. In this case, we would like to know if you would be ready to perform additional experiments (and which one(s)) to address this referee's concern?

EMBO Press usually allows one round of major revisions only, but here we would exceptionally invite a second round of revisions should you be willing to adequately address this point.

***** Reviewer's comments *****

Referee #1:

Comments on novelty/model system:

The authors did not use the best available methods to identify separate immune cell fractions. Also, the choice of the rat model systems is poorly justified.

Remarks for author:

The authors did not address one of the major concerns I had with the conclusions, for which they have provided insufficient justification.

The article's main conclusion is that Smac mimetics drive immunogenic tumour cell death and immune cell infiltration. In the current format of the manuscript there is insufficient evidence for this conclusion. The technical difficulties the authors explained as the reason for this is insufficient. For example, they have only tested one antibody for NK cells, despite a number of options available, such as dual staining for NKR-P1A and CD3, and the complement of NKR-P1B, accompanied with a high expression of CD25 and lack of CD62L, CD11b and CD27 to identify activated NK cells. Also, alternative, indirect measures of immune cell infiltration and activation could have been employed.

Additionally, the choice of the rat models is unclear. Fischer rats are immune-competent animals, further clarification is needed whether the authors used immune suppressive agents or a special sub-

strain of these rats? Similarly, please provide rationale for the choice of Hooded rats and their immunocompromised state.

Referee #2:

Comments on novelty/model system:

The authors have improved their translational study and they tried the best to also extend data to substantiate major conclusions and to use further model systems. Not all experiments or models worked, but the authors have convincingly demonstrated that technical or immunosuppressed rat model limitations due to anatomical blood flow issues prevented further evaluation. Thus, I approve the revised version, my comments were answered and addressed, manuscript in revised version fits for publication.

Remarks for author:

The study group did their best to improve the manuscript, not all worked due to anatomic limitations in rodent models, but overall one can conclude the paper was significantly improved and major conclusions strengthened.

Authors' correspondence

18th Mar 2020

Please see next page.

Key points:

- We have quantified the requested **absolute numbers** of NK and CD8+ CTLs (see below). This suggests that more CD8+ T cells are present in tumours treated with ILP-TNF/Mel/SM. Together with the enhanced induction of the activation marker Granzyme-B in such CTLs, and the tumour vaccination/re-challenge assays (Fig. 6), our data are consistent with the notion that the addition of SM to the ILP standard-of-care TNF/Mel treatment is beneficial for the treatment of soft-tissue sarcoma that arise at limb extremities.
- We apologise for not more clearly stating our rationale. Our intention was to use the **Fisher F344.Rag2^{-/-} strain** and the **Hooded pigmented Hsd:RH-Foxn1^{mu} strain**, which are immunocompromised mutants. We only used wild-type Fisher rats to evaluate the vascular anatomy of the F344 strain. For the Hooded pigmented rats, the Hsd:RH-Foxn1^{mu} strain was used. Unfortunately, neither rat strain is suitable for ILP. We have corrected our ms to clarify this point.
- Due to the current coronavirus pandemic, our animal facility has been locked down as of today, and no new experiments are permitted for time being. Therefore, we are unable to offer additional experiments. Moreover, the limited range of rat-specific antibodies further limits our capabilities to conduct additional FACS or IHCs. However, we are currently in the process of validating the anti-IFN and anti-CDC161 antibodies for IHC. If they are suitable, then we will be more than happy to include the data obtained by these antibodies. We will know more later in the week.

Below is our response to Reviewer 1, with the **reviewer 1's comments in blue boxes** and our response in 'plain text'. Modifications of the main manuscript are highlighted in **yellow (see article file)**.

Reviewer #1 (Reviewer Comments to the Author):

The major point I raised was not addressed and the conclusions were not changed accordingly.

The relative amount on NK cells and T cells increases in the tumour tissue merely due to the clearance of the dead tumour cells. This is not a proof of increased immune infiltration. For the T cell population, the altered frequency of T cell subsets confirms altered immune component in the tumour, but the reduced number of T helper cells and substantially increased number of regulatory T cells argues against immune activation. In order to corroborate the conclusion of the authors that there is increased immune infiltration and NK cell and CD8 CTL activation, additional experiments should have been carried out (either at this stage or for the studies incorporating checkpoint inhibitor targeting (CTLA4, PD1). These could have been:

For corroborating increased infiltration:

1. Determining absolute numbers of NK cells, CD8 CTLs
2. Immunohistochemistry showing infiltration

For corroborating NK and/or CD8 CTL activation:

1. Detection of induction of other activation markers, or repression of checkpoint inhibitors
2. Detection of cytokine secretion (e.g. NK cell isolation followed by IFN γ ELISA)

We have followed the reviewer's suggestion and determined absolute numbers of NK cells and CD8 CTLs (see below).

While absolute numbers of NK cells do not change upon treatment with ILP-TNF/Mel/SM, there seem to be more CD8⁺ CTLs present in tumours that were treated with ILP-TNF/Mel/SM (Appendix Fig. S4). What is more, such cytotoxic T cells also had considerably higher induction of the activation marker Granzyme B upon TNF/Mel/SM treatment. We acknowledge that the numbers are relatively small, but together with our tumour vaccination/re-challenge assays (Fig. 6), our data are consistent with the notion that the addition of SM to the ILP standard-of-care TNF/Mel treatment is beneficial for the treatment of soft-tissue sarcoma that arise at limb extremities.

We have toned down the language with regards to immune infiltration, see Abstract, Introduction Result and Discussion (see highlighted sections).

Appendix Fig. S4

IHC:

The availability of reliable, rat-specific antibodies for IHC seems to be a real problem that is difficult to address in this study. Unfortunately, there are very limited data available for the choice of rat-specific antibodies for IHCs. We have tried several, and we continue to validate additional antibodies to detect immune cells using IHCs. However, we have not been successful so far. We have tried two NK-specific surface receptors but neither antibody detected NK cells. For example, we have tried the anti-CD161 antibody on rat spleen and tumour sections. We tried at 1/100 with both high and low pH retrieval but the only positive signal we are seeing is in the cytoplasm of plasma cells, which is quite strong using both treatments and in both tissues. This is in line with the poor (1*) review posted on the website for this antibody using IHC on FFPE tissues, for which the antibody is not indicated by the manufacturer.

We will update this reviewer should we identify antibodies that reliably detect NK cells or activation markers.

Additional *in vivo* experiments and coronavirus pandemic:

The current coronavirus pandemic made it necessary for our institute to close the animal facility as of today. Therefore, additional animal experiments are no longer possible. We sincerely apologise for this situation, which is beyond our control. It remains unclear when additional *in vivo* experiments might be possible.

2nd Editorial Decision

22nd Mar 2020

Thank you for the submission of your manuscript to EMBO Molecular Medicine, and for providing further revision to address referee #1's remaining concerns.

You will see from the enclosed reports that both reviewers are now supportive of publication, and I am thus pleased to inform you that we will be able to accept your manuscript pending the following final editorial amendments.

***** Reviewer's comments *****

Referee #1 (Remarks for Author):

The authors have addressed the key points I raised and with the Coronavirus lab close-downs it would be unrealistic to request further experiments. As the conclusions in the MS are toned down and absolute cell counts have been included, I am happy to approve the MS for publication.

Referee #2 (Comments on Novelty/Model System for Author):

The authors have improved their translational study and they tried their best to also extend data to substantiate major conclusions and to use further model systems. Not all experiments or models worked, but the authors have convincingly demonstrated that technical or immunosuppressed rat model limitations due to anatomical blood flow issues prevented further evaluation. Thus, I approve the revised version, my comments were answered and addressed, manuscript in revised version fits for publication.

Referee #2 (Remarks for Author):

The study group did their best to improve the manuscript, not all worked due to anatomic limitations in rodent models, but overall one can conclude the paper was significantly improved and major conclusions strengthened.

2nd Revision - authors' response

15th Apr 2020

The authors performed the requested editorial changes.

Corresponding Author Name: Pascal Meier

Journal Submitted to: EMBO MOLECULAR MEDICINE

Manuscript Number: EMM-2019-10979-V2